# A Black-Box Debiasing Framework for Conditional Sampling

**Han Cui**
University of Illinois at Urbana-Champaign
Champaign, IL
hancui5@illinois.edu

**Jingbo Liu**
University of Illinois at Urbana-Champaign
Champaign, IL
jingbol@illinois.edu

## Abstract

Conditional sampling is a fundamental task in Bayesian statistics and generative modeling. Consider the problem of sampling from the posterior distribution $P_{X|Y=y^*}$ for some observation $y^*$, where the likelihood $P_{Y|X}$ is known, and we are given $n$ i.i.d. samples $D = \{X_i\}_{i=1}^n$ drawn from an unknown prior distribution $\pi_X$. Suppose that $f(\hat{\pi}_{X^n})$ is the distribution of a posterior sample generated by an algorithm (e.g. a conditional generative model or the Bayes rule) when $\hat{\pi}_{X^n}$ is the empirical distribution of the training data. Although averaging over the randomness of the training data $D$, we have $\mathbb{E}_D(\hat{\pi}_{X^n}) = \pi_X$, we do not have $\mathbb{E}_D\{f(\hat{\pi}_{X^n})\} = f(\pi_X)$ due to the nonlinearity of $f$, leading to a bias. In this paper we propose a black-box debiasing scheme that improves the accuracy of such a naive plug-in approach. For any integer $k$ and under boundedness of the likelihood and smoothness of $f$, we generate samples $\hat{X}^{(1)}, \ldots, \hat{X}^{(k)}$ and weights $w_1, \ldots, w_k$ such that $\sum_{i=1}^k w_i P_{\hat{X}^{(i)}}$ is a $k$-th order approximation of $f(\pi_X)$, where the generation process treats $f$ as a black-box. Our generation process achieves higher accuracy when averaged over the randomness of the training data, without degrading the variance, which can be interpreted as improving memorization without compromising generalization in generative models.

## 1 Introduction

Conditional sampling is a major task in Bayesian statistics and generative modeling. Given an observation $y^*$, the objective is to draw samples from the posterior distribution $P_{X|Y=y^*}$, where the likelihood $P_{Y|X}$ is known but the prior distribution $\pi_X$ is unknown. Instead, we are provided with a dataset $D = \{X_i\}_{i=1}^n$ consisting of $n$ i.i.d. samples drawn from $\pi_X$.

The setting is common in a wide range of applications, including inpainting and image deblurring [9, 5] (where $X$ is an image and $Y|X$ is a noisy linear transform), text-conditioned image generation [7, 13](where $X$ is an image and $Y$ is a natural language prompt), simulating biomedical structures with desired properties, and trajectory simulations for self-driving cars. Moreover, conditional sampling is equally vital in high-impact machine learning and Bayesian statistical methods, particularly under distribution shift, such as in transfer learning. For instance, conditional sampling has enabled diffusion models to generate trajectories under updated policies, achieving state-of-the-art performance in offline reinforcement learning [8, 1, 26]. Pseudo-labeling, a key technique for unsupervised pretraining [10] and transfer learning calibration [20], relies on generating conditional labels. Additionally, conditional diffusion models seamlessly integrate with likelihood-free inference [6, 18, 27]. Existing approaches often use generative models such as *VAEs* or *Diffusion models* to generate samples by learning $P_{X|Y=y^*}$ implicitly from the data.

39th Conference on Neural Information Processing Systems (NeurIPS 2025).

Our work focuses on approximating the true posterior $P_{X|Y=y^*}$ using the observed samples $D = X^n = (X_1, \ldots, X_n)$ and the new observation $y^*$, but without the knowledge of the prior. Denote by $P_{\hat{X}|Y=y^*,D}$ the approximating distribution. We can distinguish two kinds of approximations: First, $P_{\hat{X}|Y=y^*,D} \approx P_{X|Y=y^*}$ with high probability over $D$, which captures the *generalization* ability since the model must learn the distribution from the training samples. This criterion is commonly adopted in estimation theory and has also been examined in the convergence analysis of generative models [16, 28, 26, 22]. Second, $\mathbb{E}_D(P_{\hat{X}|Y=y^*,D}) \approx P_{X|Y=y^*}$ is a weaker condition since it only requires approximation when averaged over the randomness of the training data, but is still useful in some sampling and generative tasks, e.g. generating samples for bootstrapping or Monte Carlo estimates of function expectations. The second condition captures the ability to *memorize* or *imitate* training sample distribution. It is interesting to note that in the unconditional setting (i.e., without distribution shift), a permutation sampler can perfectly imitate the unknown training data distribution, even if $n = 1$, so the problem is trivial from the sample complexity perspective. However, in the conditional setting, it is impossible to get such a perfect imitation with finite training data, as a simple binary distribution example in Section 3.2 illustrates. It naturally leads to the following question:

*How fast can the posterior approximation converge to the true posterior as $n \to \infty$, and is there a sampling scheme achieving this convergence rate?*

**Contribution.** We address the question above by proposing a novel debiasing framework for posterior approximation. Our main contributions can be summarized as follows:

- **Debiasing framework for posterior approximation.** We introduce a novel debiasing framework for posterior approximation with an unknown prior. Our method leverages the known likelihood $P_{Y|X}$ and the observed data to construct an improved approximate posterior $\widetilde{P}_{X^n}(x|y^*)$ with provably reduced bias. In particular, let $f(\hat{\pi}_{X^n})$ represent the distribution of a posterior sample generated by an algorithm $f$ when $\hat{\pi}_{X^n}$ is the empirical distribution of the training data. Then for any integer $k$, assuming that the likelihood function $P_{Y|X}$ is bounded and $f$ is sufficiently smooth, we generate samples $\hat{X}^{(1)}, \ldots, \hat{X}^{(k)}$ from $f$ based on multiple resampled empirical distributions. These are then combined with designed (possibility negative) weights $w_1, \ldots, w_k$ to construct an approximate posterior:

$$\widetilde{P}_{X^n}(\cdot|y^*) = \sum_{i=1}^{k} w_i P_{\hat{X}^{(i)}}$$

which is a $k$-th order approximation of $f(\pi_X)$, treating the generation process $f$ as a black-box. Our generation process achieves higher accuracy when averaged over the randomness of the training data, but not conditionally on the training data, which highlights the trade-off between memorization and generalization in generative models. Specifically, we do not assume any parametric form for the prior and our method can achieve a bias rate of $\mathcal{O}(n^{-k})$ for any prescribed integer $k$ and a variance rate of $\mathcal{O}(n^{-1})$.

- **Theoretical bias and variance guarantees.** We establish theoretical guarantees on both bias and variance for the Bayes-optimal sampler under continuous prior setting and for a broad class of samplers $f$ with a continuous $2k$-th derivative, as specified in Assumption 2, under the discrete prior setting. The proposed debiasing framework can also be applied in a black-box manner (see Remark 2 for the intuition), making it applicable to a broad class of state-of-the-art conditional samplers, such as diffusion models and conditional VAE. Based on this perspective, we treat the generative model $f$ as a black box that can output posterior samples given resampled empirical distributions. Applying $f$ to multiple recursive resampled versions of the training data and combining the outputs with polynomial weights, we obtain a bias-corrected approximation of the posterior. The procedure is described in Algorithm 1.

Our approach is also related to importance sampling. Since the true posterior $P_{X|Y}$ is intractable to compute, we can use expectations under the debiased posterior $\widetilde{P}_{X^n}(x|y^*)$ to approximate the expectations under the true posterior $P_{X|Y=y^*}$. For a test function $h$, we estimate $\mathbb{E}_{P_{X|Y=y^*}}\{h(X)\}$ by

$$\mathbb{E}_{\widetilde{P}_{X^n}(x|y^*)}\{h(X)\} \approx \frac{1}{N} \sum_{i=1}^{N} h(\tilde{X}_j) \frac{\widetilde{P}_{X^n}(\tilde{X}_j|y^*)}{q(\tilde{X}_j|y^*)}, \tag{1}$$

---

**Algorithm 1** Posterior Approximation via Debiasing Framework

---

**Input:** Observation $y^*$, likelihood $P_{Y|X}$, data $X^n = (X_1, \ldots, X_n)$, number of steps $k$, a black-box conditional sampler $f$ (i.e., a map from a prior distribution to a posterior distribution)

**Output:** $\hat{X}^{(j)}, j = 1, \ldots, k$ such that $\sum_{j=0}^{k-1} \binom{k}{j+1}(-1)^j P_{\hat{X}^{(j+1)}}$ is a high-order approximation of the posterior $P_{X|Y=y^*}$

 1: Initialize $\hat{p}^{(1)} := \hat{\pi}_{X^n}$
 2: **for** $\ell = 2$ to $k$ **do**
 3:     Generate $n$ i.i.d. samples from $\hat{p}^{(\ell-1)}$
 4:     Let $\hat{p}^{(\ell)}$ be the empirical distribution of the sampled data
 5: **end for**
 6: **for** $j = 1$ to $k$ **do**
 7:     Generate samples $\hat{X}^{(j)} \sim f(\hat{p}^{(j)})$
 8: **end for**
 9: Return $\hat{X}^{(j)}, j = 1, \ldots, k$

---

where $\tilde{X}_j \sim q(x|y^*)$ for a chosen proposal distribution $q$. This resembles our method, in which we approximate the true posterior by a weighted combination $\sum_{i=1}^{k} w_i P_{\hat{X}^{(i)}}$. And in (1), the term $\widetilde{P}_{X^n}(\tilde{X}_j|y^*)/q(\tilde{X}_j|y^*)$ can be interpreted as a weight assigned to each sample, analogous to the weights $w_i$ in our framework. Therefore, we expect that Algorithm 1 can be broadly applied to Monte Carlo estimates of function expectations, similar to the standard importance sampling technique.

## 2   Related work

**Jackknife Technique.** Our work is related to the jackknife technique [17], a classical method for bias reduction in statistical estimation that linearly combines estimators computed on subsampled datasets. Specifically, the jackknife technique generates leave-one-out (or more generally, leave-$s$-out where $s \geq 1$) versions of an estimator, and then forms a weighted combination to cancel lower-order bias terms. Recently, Nowozin [14] applied the jackknife to the importance-weighted autoencoder (IWAE) bound $\hat{\mathcal{L}}_n$, which estimates the marginal likelihood $\log \pi(x)$ using $n$ samples. While $\hat{\mathcal{L}}_n$ is proven to be an estimator with bias of order $\mathcal{O}(n^{-1})$, the jackknife correction produces a new estimator with reduced bias of order $\mathcal{O}(n^{-m})$. Our paper introduces a debiaisng framework based on the similar idea that using a linear combination of multiple approximations to approximate the posterior.

**Conditional Generative Models.** Conditional generative models have become influential and have been extensively studied for their ability to generate samples from the conditional data distribution $P(\cdot|y)$ where $y$ is the conditional information. This framework is widely applied in vision generation tasks such as text-to-image synthesis [13, 24, 2] where $y$ is an input text prompt, and image inpainting [11, 21] where $y$ corresponds to the known part of an image. We expect that our proposed debiasing framework could work for a broad class of conditional generative models to construct a high order approximation of the posterior $P(\cdot|y)$.

**Memorization in Generative Models.** The trade-off between memorization and generalization has been a focus of research in recent years. In problems where generating new structures or preserving privacy of training data is of high priority, generalization is preferred over memorization. For example, a study by Carlini et al. [4] demonstrates that diffusion models can unintentionally memorize specific images from their training data and reproduce them when generating new samples. To reduce the memorization of the training data, Somepalli et al. [19] applies randomization and augmentation techniques to the training image captions. Additionally, Yoon et al. [25] investigates the connection between generalization and memorization, proposing that these two aspects are mutually exclusive. Their experiments suggest that diffusion models are more likely to generalize when they fail to memorize the training data. On the other hand, memorizing and imitating the training data may be intentionally exploited, if the goal is Monte Carlo sampling for evaluations of expected values, or if the task does not involve privacy issues, e.g. image inpainting and reconstruction. In these applications, the ability to imitate or memorize the empirical distribution of the training data becomes essential, especially when generalization is unattainable due to the insufficient data. Our work focuses

on memorization phase and shows that it is possible to construct posterior approximations with provably reduced bias by exploiting the empirical distribution.

**Mixture-Based Approximation of Target Distributions.** Sampling from a mixture of distributions $a_1 P_{X_1} + a_2 P_{X_2} + \cdots + a_k P_{X_k}$ to approximate a target distribution $P^*$ is commonly used in Bayesian statistics, machine learning, and statistical physics, especially when individual samples or proposals are poor approximations, but their ensemble is accurate. Traditional importance sampling methods often rely on positive weights, but recent work has expanded the landscape to include more flexible and powerful strategies, including the use of signed weights and gradient information. For example, Oates et al. [15] uses importance sampling and control functional estimators to construct a linear combination of estimators with weights $a_k$ to form a variance-reduced estimator for an expectation under a target distribution $P^*$. Liu and Lee [12] select the weights $a_k$ by minimizing the empirical version of the kernelized Stein discrepancy (KSD), which often results in negative weights.

# 3 Problem setup and notation

Consider a dataset $\{X_i\}_{i=1}^n$ consisting of $n$ independent and identically distributed (i.i.d.) samples, where $X_i \in \mathcal{X}$ is drawn from an unknown prior distribution $\pi_X$ and the conditional distribution $P_{Y|X}$ is assumed to be known. In the Bayesian framework, the posterior distribution of $X$ given $Y$ is given by

$$P_{X|Y}(dx|y) = \frac{P_{Y|X}(y|x)\pi_X(dx)}{\int P_{Y|X}(y|x)\pi_X(dx)}.$$

Given the observed data $X^n = (X_1, \cdots, X_n)$ and the new observation $y^*$, our goal is to approximate the true posterior $P_{X|Y=y^*}$.

## 3.1 Naive plug-in approximation

A natural approach is to replace the unknown prior $\pi_X$ with its empirical counterpart

$$\hat{\pi}_{X^n} = n^{-1} \sum_{i=1}^n \delta_{X_i}$$

in the Bayes' rule to compute an approximate posterior which yields the plug-in posterior

$$\widehat{P}_{X|Y}(dx|y^*) = \frac{P_{Y|X}(y^*|x)\hat{\pi}_{X^n}(dx)}{\int P_{Y|X}(y^*|x)\hat{\pi}_{X^n}(dx)}. \tag{2}$$

Note that even though $\mathbb{E}_D(\hat{\pi}_{X^n}) = \pi_X$, the nonlinearity of Bayes' rule makes the resulting posterior (2) still biased, that is, $\mathbb{E}_D\left\{\widehat{P}_{X|Y}(\cdot|y^*)\right\} \neq P_{X|Y}(\cdot|y^*)$. If the denominator in (2) were replaced with $\int P_{Y|X}(y^*|x)\pi_X(dx)$, then averaging the R.H.S. of (2) over the randomness in $X^n$ would yield the true posterior $P_{X|Y}(dx|y^*) = P_{Y|X}(y^*|x)\pi_X(dx)/\int P_{Y|X}(y^*|x)\pi_X(dx)$ exactly.

For typical choices of $P_{Y|X}$ which have nice conditional density (e.g., the additive Gaussian noise channel), $\int P_{Y|X}(y^*|x)\hat{\pi}_{X^n}(dx)$ converges at the rate of $n^{-1/2}$, by the central limit theorem. Consequently, $\mathbb{E}_D(\widehat{P}_{X|Y=y^*})$ converges to the true posterior at the rate $\tilde{\mathcal{O}}(n^{-1/2})$ in the $\infty$-Renyi divergence metric regardless of the smoothness of $\pi_X$. Under appropriate regularity conditions, we can in fact show that $\mathbb{E}_D(\widehat{P}_{X|Y=y^*})$ converges at the rate of $\tilde{\mathcal{O}}(n^{-1})$, which comes from the variance term in the Taylor expansion. Naturally, we come to an essential question: can we eliminate the bias entirely? That is, is it possible that $\mathbb{E}_D\{\widehat{P}_{X|Y}(\cdot|y^*)\} = P_{X|Y}(\cdot|y^*)$?

## 3.2 Impossibility of exact unbiasedness

Exact unbiasedness is, in general, unattainable. Consider the simple case where $X$ is binary, that is, $X \sim \text{Bern}(q)$ for some unknown parameter $q \in (0, 1)$. Define the likelihood ratio $\alpha = \alpha(y^*) :=$

$P_{Y|X}(y^*|1)/P_{Y|X}(y^*|0)$. Then the true posterior is

$$X|Y = y^* \sim \text{Bern}\left(\frac{\alpha q}{\alpha q + 1 - q}\right).$$

On the other hand, if we approximate the posterior distribution as $\hat{P}_{X|Y}(1|y = y^*) = \text{Bern}(p(k))$ upon seeing $k$ outcomes equal to 1, then

$$\mathbb{E}_D\left\{\hat{P}_{X|Y}(1|y^*)\right\} = \sum_{k=0}^{n} p(k)\binom{n}{k}q^k(1-q)^{n-k}, \tag{3}$$

which is a polynomial function of $q$, and hence cannot equal the rational function $\alpha q/(\alpha q + 1 - q)$ for all $q$. This implies that an *exact* imitation, in the sense that $\mathbb{E}_D\{\hat{P}_{X|Y}(\cdot|y^*)\} = P_{X|Y}(\cdot|y^*), \forall \pi_X$, is impossible. However, since a rational function can be *approximated* arbitrarily well by polynomials, this does not rule out the possibility that a better sampler achieving convergence faster than, say, the $\tilde{\mathcal{O}}(n^{-1/2})$ rate of the naive plug-in method. Indeed, in this paper we propose a black-box method that can achieve convergence rates as fast as $\mathcal{O}(n^{-k})$ for any fixed $k > 0$.

### 3.3 Objective and notation

Since the bias in the plug-in approximation arises from the nonlinearity of Bayes' rule, we aim to investigate whether a faster convergence rate can be achieved. Our objective is to construct an approximation $\widetilde{P}_{X^n}(x|y = y^*)$ that improves the plug-in approximation by reducing the bias. Specifically, the debiased approximation satisfies the following condition:

$$\left|\mathbb{E}_{X^n}\left\{\widetilde{P}_{X^n}(x|y = y^*)\right\} - P_{X|Y}(x|y^*)\right| < \left|\mathbb{E}_{X^n}\left\{\hat{P}_{X|Y}(x|y^*)\right\} - P_{X|Y}(x|y^*)\right|.$$

More generally, we can replace the Bayes rule by an arbitrary map $f$ from a prior to a posterior distribution (e.g. by a generative model), and the goal is a construct a debiased map $\tilde{f}$ such that

$$\left\|\mathbb{E}_{X^n}\tilde{f}(\hat{\pi}_{X^n}) - f(\pi)\right\|_{\text{TV}} < \left\|\mathbb{E}_{X^n}f(\hat{\pi}_{X^n}) - f(\pi)\right\|_{\text{TV}}.$$

**Notation.** Let $\delta_x$ denote the Dirac measure, $\|\cdot\|_{\text{TV}}$ denote the total variation norm. For any positive integer $m$, denote $[m] = \{1, \ldots, m\}$ as the set of all positive integers smaller than all equal to $m$. Write $b_n = \mathcal{O}(a_n)$ if $b_n/a_n$ is bounded as $n \to \infty$. Write $b_n = \mathcal{O}_s(a_n)$ if $b_n/a_n$ is bounded by $C(s)$ as $n \to \infty$ for some constant $C(s)$ that depends only on $s$. We use the notation $a \lesssim b$ to indicate that there exists a constant $C > 0$ such that $a \leq Cb$. Similarly, $a \lesssim_k b$ means that there exists a constant $C(k) > 0$ that depends only on $k$ such that $a \leq C(k)b$. Furthermore, for notational simplicity, we will use $\pi$ to denote the true prior $\pi_X$ and $\hat{\pi}$ to denote the empirical prior $\hat{\pi}_{X^n}$ in the rest of the paper.

## 4 Main result

### 4.1 Debiased posterior approximation under continuous prior

Let $\Delta_{\mathcal{X}}$ denote the space of probability measures on $\mathcal{X}$. Define the likelihood function $\ell(x) := P_{Y|X}(y^*|x)$, which represents the probability of observing the data $y^*$ given $x$. Let $f : \Delta_{\mathcal{X}} \to \Delta_{\mathcal{X}}$ be a map from the prior measure to the posterior measure, conditioned on the observed data $y^*$. Let $B_n$ be the operator such that for any function $f : \Delta_{\mathcal{X}} \to \Delta_{\mathcal{X}}$,

$$B_n f(p) = \mathbb{E}\left\{f(\hat{p})\right\}, \tag{4}$$

where $\hat{p}$ denotes the empirical measure of $n$ i.i.d. samples from measure $p$.

We consider the case that $f$ represents a mapping corresponding to the Bayes posterior distribution. Using Bayes' theorem, for any measure $\pi \in \Delta_{\mathcal{X}}$ and any measurable set $A \subset \mathcal{X}$, the posterior measure $f(\pi)$ is expressed as

$$f(\pi)(A) = \frac{\displaystyle\int_A \ell(x)\pi(dx)}{\displaystyle\int_{\mathcal{X}} \ell(x)\pi(dx)}.$$

As discussed in Section 3, the equality $B_n f(\pi) = f(\pi)$ is not possible due to the nonlinearity of $f$. However, we can achieve substantial improvements over the plug-in method by using polynomial approximation techniques analogous to those from prior statistical work by Cai and Low [3] and Wu and Yang [23]. For $k \geq 1$, we define the operator $D_{n,k}$ as a linear combination of the iterated operators $B_n^j$ for $j = 0, \ldots, k-1$:

$$D_{n,k} = \sum_{j=0}^{k-1} \binom{k}{j+1} (-1)^j B_n^j.$$

**Assumption 1.** *The likelihood function $\ell$ is bounded, i.e., there exists $0 < L_1 \leq L_2$ such that $L_1 \leq \ell(x) \leq L_2$.*

The following theorem provides a systematic method for constructing an approximation of $f(\pi)$ with an approximation error of order $\mathcal{O}(n^{-k})$ for any desired integer $k$.

**Theorem 1.** *Under Assumption 1, for any measurable set $A \subset \mathcal{X}$ and any $k \in \mathbb{N}^+$, we have*

$$\|\mathbb{E}_{X^n} \{D_{n,k} f(\hat{\pi})\} - f(\pi)\|_{\mathrm{TV}} = \mathcal{O}_{L_1, L_2, k}(n^{-k}), \tag{5}$$

$$\mathrm{Var}_{X^n} \{D_{n,k} f(\hat{\pi})(A)\} = \mathcal{O}_{L_1, L_2, k}(n^{-1}). \tag{6}$$

**Remark 1.** $D_{n,k} f(\hat{\pi}) = \sum_{j=0}^{k-1} \binom{k}{j+1}(-1)^j B_n^j f(\hat{\pi})$ *in (5) can be interpreted as a weighted average of the distribution of some samples. Specifically, if we treat the coefficient $\binom{k}{j+1}(-1)^j$ as the weight $w_j$ and $B_n^j f(\hat{\pi})$ as the distribution of some sample $\hat{X}^{(j)}$, then $D_{n,k} f(\hat{\pi}) = \sum_{j=0}^{k-1} w_j P_{\hat{X}^{(j)}}$.*

**Remark 2.** *Recall the binary case discussed in Section 3, (3) illustrates that we cannot get an exact approximation for the true posterior. But from (5), we demonstrate that even if $\|\mathbb{E}_{X^n}\{D_{n,k}f(\hat{\pi})\} - f(\pi)\|_{\mathrm{TV}} = 0$ is impossible, it can be arbitrarily small. Although the theoretical guarantees are derived for the Bayes-optimal sampler, (5) is expected to hold for general sampler $f$ such as diffusion models. Here we give the intuition for this conjecture. We view the operator $B_n f(\pi) := \mathbb{E}\{f(\hat{\pi})\}$ as a good approximation of $f(\pi)$, i.e., $B_n \approx I$, where $I$ is the identity operator. This implies that the error operator $E := I - B_n$ is a "small" operator. Under this heuristic, if $E f(\pi) = \mathcal{O}(n^{-1})$, intuitively we have $E^k f(\pi) = \mathcal{O}(n^{-k})$. Using the binomial expansion of $E^k = (I - B_n)^k$, we have $E^k f(\pi) = f(\pi) - \sum_{j=1}^{k} \binom{k}{j}(-1)^{j-1} B_n^j f(\pi) = f(\pi) - \mathbb{E}\{\sum_{j=1}^{k} \binom{k}{j}(-1)^{j-1} B_n^{j-1} f(\hat{\pi})\} = f(\pi) - \mathbb{E}\{D_{n,k} f(\hat{\pi})\}$. This representation motivates the specific form of $D_{n,k}$.*

**Remark 3.** *In general, the curse of dimensionality may arise and depends on the specific distribution of $X$ and the likelihood function $\ell$. There is no universal relationship between $n$ and the dimension $d$. However, to build intuition, we give an example that illustrates how $n$ and $d$ may relate. Suppose that $Y = \big(Y(1), \ldots, Y(d)\big)$ and $X = \big(X(1), \ldots, X(d)\big)$ have i.i.d. components, and $L_1 \leq P\big(Y(i)|X(i)\big) \leq L_2$ for $1 \leq i \leq d$. Then we have $\ell(X) := P(Y|X) \in [L_1^d, L_2^d]$. Note that $\mathcal{O}_{L_1, L_2, k}(n^{-k})$ in (5) can be bounded by $C(k)(L_2^d/L_1^d)^{2k} n^{-k}$ for some constant $C(k)$ depending only on $k$. To ensure that our debiasing method improves over the baseline method without debiasing in the case of growing dimensions, it suffices to let $n$ and $d$ satisfy that $(L_2^d/L_1^d)^{2k} n^{-k} \ll n^{-1}$ when $k \geq 2$, which is equivalent to $kd \ll \log(n)$.*

***Sketch proof for Theorem 1.*** First let $\mu = \int_{\mathcal{X}} \ell(x)\pi(dx)$ and $\mu_A = \int_A \ell(x)\pi(dx)$ and introduce a new operator

$$C_{n,k} := \sum_{j=1}^{k} \binom{k}{j}(-1)^{j-1} B_n^j,$$

then we have $B_n D_{n,k} = C_{n,k}$. By the definition of $B_n$, it suffices to show that

$$C_{n,k} f(\pi)(A) - f(\pi)(A) = \mathcal{O}_{L_1, L_2, k}(n^{-k}).$$

The first step is to express $B_n^j f(\pi)$ with the recursive resampled versions of the training data. Specifically, let $\hat{\pi}^{(0)} = \pi, \hat{\pi}^{(1)} = \hat{\pi}$ and set $(X_1^{(0)}, \ldots, X_n^{(0)}) \equiv (X_1, \ldots, X_n)$. For $j = 1, \ldots, k,$

we define $\hat{\pi}^{(j)}$ as the empirical measure of $n$ i.i.d. samples $(X_1^{(j-1)}, \ldots, X_n^{(j-1)})$ drawn from the measure $\hat{\pi}^{(j-1)}$. Additionally, let

$$e_n^{(j)} = n^{-1} \sum_{i=1}^{n} \left\{ \ell(X_i^{(j)}) - \mu \right\} \quad \text{and} \quad \mu_A^{(j)} = n^{-1} \sum_{i=1}^{n} \ell(X_i^{(j)}) \delta_{X_i^{(j)}}(A).$$

Then we have

$$C_{n,k} f(\pi)(A) = \sum_{j=1}^{k} \binom{k}{j} (-1)^{j-1} B_n^j f(\pi)(A) = \sum_{j=1}^{k} \binom{k}{j} (-1)^{j-1} \mathbb{E} \left( \frac{\mu_A^{(j-1)}}{e_n^{(j-1)} + \mu} \right). \quad (7)$$

The second step is to rewrite (7) with Taylor expansion of $\mu_A^{(j-1)}/(e_n^{(j-1)} + \mu)$ with respect to $e_n^{(j-1)}$ up to order $2k - 1$. $L_1 \leq l(X_i^{(j-1)}) \leq L_2$ and Hoeffding's inequality implies that the expectation of the residual term $\mathbb{E}\{(e_n^{(j-1)})^{2k}/\xi^{2k+1}\}$ for some $\xi$ between $e_n^{(j-1)} + \mu$ and $\mu$ is $\mathcal{O}_{L_1, L_2, k}(n^{-k})$. Now we instead to show that

$$B_{k,r} := \sum_{j=1}^{k} \binom{k}{j} (-1)^{j-1} \mathbb{E} \left\{ \mu_A^{(j-1)} (e_n^{(j-1)})^r \right\} = \mathcal{O}_{L_1, L_2, k}(n^{-k}),$$

since (7) is equal to $\mu_A/\mu + \sum_{r=1}^{2k-1} (-1)^r \mu^{-r-1} B_{k,r} + \mathcal{O}_{L_1, L_2, k}(n^{-k})$.

Define a new operator $B : h \mapsto \mathbb{E}[h(\hat{\pi})]$ for any $h : \Delta_{\mathcal{X}} \to \mathbb{R}$ and let $h_s(\pi) = \{\int_A \ell(x)\pi(dx)\}\{\int \ell(x)\pi(dx)\}^s$. Then

$$B_{k,r} = \sum_{s=0}^{r} \binom{r}{s} (-1)^{(r-s)} \mu^{r-s} \sum_{j=1}^{k} \binom{k}{j} (-1)^{j-1} B^j h_s(\pi).$$

The last step is to prove

$$(I - B)^k h_s(\pi) = \mathcal{O}_{L_1, L_2, s}(n^{-k}), \quad (8)$$

since (8) is equivalent to $\sum_{j=1}^{k} \binom{k}{j} (-1)^{j-1} B^j h_s(\pi) = h_s(\pi) + \mathcal{O}_{L_1, L_2, s}(n^{-k})$. Finally (8) follows from the fact that $(I - B)^k h_s(\pi)$ can be expressed as a finite sum of the terms which have the following form:

$$\alpha_{\mathbf{a}, \mathbf{s}, v} \left\{ \int_A \ell^v(x)\pi(dx) \right\} \prod_i \left\{ \int \ell^{a_i}(x)\,\pi(dx) \right\}^{s_i},$$

where $|\alpha_{\mathbf{a}, \mathbf{s}, v}| \leq C_k(s) n^{-k}$ for some constnat $C_k(s)$ (see our Lemma 2).

$\square$

## 4.2 Debiased posterior approximation under discrete prior

In this section, we consider the case where $X$ follows a discrete distribution. As mentioned in Remark 2, the result in Theorem 1 is expected to hold in a broader class of samplers $f$ under smoothness, extending beyond just the Bayes-optimal sampler $f$. The assumption of finite $\mathcal{X}$ in this section allows us to simplify some technical aspects in the proof.

Let the support of $X$ be denoted as $\mathcal{X} = \{u_1, u_2, \ldots, u_m\}$. Assume that $|\mathcal{X}| = m$ is finite, and $X$ is distributed according to an unknown prior distribution $\pi(x)$ such that the probability of $X$ taking the value $u_i$ is given by $\pi(X = u_i) = q_i$ for $i = 1, 2, \ldots, m$. Here, the probabilities $q_i$ are unknown and satisfy the usual constraints that $q_i \geq 0$ for all $i$ and $\sum_{i=1}^{m} q_i = 1$.

Let $\mathbf{q} = (q_1, \cdots, q_m)^\top$ represent the true prior probability vector associated with the probability distribution $\pi(x)$. Let $\mathbf{g}$ be a map from a prior probability vector to a posterior probability vector. Then $\mathbf{g}(\mathbf{q}) = (g_1(\mathbf{q}), \cdots, g_m(\mathbf{q}))^\top$ is the probability vector associated with the posterior. Let $\mathbf{T} = (T_1, \cdots, T_m)^\top$ where $T_j = \sum_{i=1}^{n} \mathbb{1}_{X_i = u_j}$ for $j = 1, \cdots, m$. In such setting, by the definition (4) of operator $B_n$, we can rewrite the operator $B_n$ as

$$B_n g_s(\mathbf{q}) = \mathbb{E}\left\{ g_s(\mathbf{T}/n) \right\} = \sum_{\boldsymbol{\nu} \in \bar{\Delta}_m} g_s\left(\frac{\boldsymbol{\nu}}{n}\right) \binom{n}{\boldsymbol{\nu}} \mathbf{q}^{\boldsymbol{\nu}},$$

where $\bar{\Delta}_m = \{\boldsymbol{\nu} \in \mathbb{N}^m : \sum_{j=1}^m \nu_j = n\}$ and

$$\binom{n}{\boldsymbol{\nu}} = \frac{n!}{\nu_1! \cdots \nu_m!}, \quad \mathbf{q}^{\boldsymbol{\nu}} = q_1^{\nu_1} \cdots q_m^{\nu_m}.$$

Additionally, let $\Delta_m = \{\mathbf{q} \in \mathbb{R}^m : q_j \geq 0, \sum_{j=1}^m q_j = 1\}$ and let $\|\cdot\|_{C^k(\Delta_m)}$ denote the $C^k(\Delta_m)$-norm which is defined as $\|f\|_{C^k(\Delta_m)} = \sum_{\|\boldsymbol{\alpha}\|_1 \leq k} \|\partial^{\boldsymbol{\alpha}} f\|_\infty$ for any $f \in C^k(\Delta_m)$.

**Assumption 2.** $|\mathcal{X}| = m$ *is finite, and* $\max_{s \in [m]} \|g_s\|_{C^{2k}(\Delta_m)} \leq G$ *for some constant G.*

The following theorem provides a systematic method for constructing an approximation of $g_s(\mathbf{q})$ with an error of order $\mathcal{O}(n^{-k})$ for any desired integer $k$.

**Theorem 2.** *If* $|\mathcal{X}| = m$, *let* $\mathbf{q} = (q_1, \cdots, q_m)^\top$ *be the true prior probability vector associated with a discrete probability distribution and* $\mathbf{T} = (T_1, \cdots, T_m)^\top$ *where* $T_j = \sum_{i=1}^n \mathbb{1}_{X_i = u_j}$ *for* $j = 1, \cdots, m$. *Under Assumption 2, the following holds for any* $s \in \{1, \cdots, m\}$ *and any* $k \in \mathbb{N}^+$:

$$\mathbb{E}_{X^n} \{D_{n,k}(g_s)(\mathbf{T}/n)\} - g_s(\mathbf{q}) = \mathcal{O}_{k,m,G}(n^{-k}),$$
$$\mathrm{Var}_{X^n} \{D_{n,k}(g_s)(\mathbf{T}/n)\} = \mathcal{O}_{k,m,G}(n^{-1}).$$

Theorem 2 follows directly from the following lemma, which provides the key approximation result.

**Lemma 1.** *For any integers* $k, m \in \mathbb{N}^+$ *and any function* $f \in C^k(\Delta_m)$, *we have*

$$\|C_{n,\lceil k/2 \rceil}(f) - f\|_\infty = \|(B_n - I)^{\lceil k/2 \rceil}(f)\|_\infty \lesssim_{k,m} \|f\|_{C^k(\Delta_m)} n^{-k/2}.$$

Note that Theorem 2 holds for all mappings $\mathbf{g}$ that satisfy Assumption 2. When $\mathbf{g}$ represents a mapping corresponding to the Bayes posterior distribution, we know the exact form of $\mathbf{g}(\mathbf{q})$. Hence, we can explore sampling schemes for Bayes-optimal mapping $\mathbf{g}$.

We claim that Bayes-optimal mapping $\mathbf{g}$ satisfies Assumption 2. In fact, let $\ell_s = \ell(u_s) := P_{Y|X}(y^*|u_s)$. Using Bayes' theorem, the posterior probability of $X = u_s$ given $y^*$ is given by

$$P_{X|Y}(u_s|y^*) = \frac{\ell_s q_s}{\sum_{j=1}^m \ell_j q_j}.$$

In this case, $g_s(\mathbf{q}) := \ell_s q_s / \sum_{j=1}^m \ell_j q_j$ for $s = 1, \cdots, m$. Since $|\mathcal{X}| = m$ is finite, we know that there exists a constant $c_1, c_2 > 0$ such that $c_1 \leq l_j \leq c_2$ for all $1 \leq j \leq m$, which implies that $\max_{s \in [m]} \|g_s\|_{C^{2k}(\Delta_m)} \leq G$ for some constant $G$ based on $k$.

Moreover, estimating $g_s(\mathbf{q})$ based on the observations of $X^n = (X_1, \cdots, X_n)$ and $y^*$ is sufficient to generate samples from the posterior distribution $P_{X|Y}(u_s|y^*)$ for $s = 1, \cdots, m$. Since the exact form of $g_s$ is known, if we let $\widetilde{P}_{X^n}(x = u_s|y = y^*) = D_{n,k}(g_s)(\mathbf{T}/n)$ where $\mathbf{T}/n$ denotes the empirical of the training set, we obtain the following theorem.

**Theorem 3.** *Under Assumption 2, for any* $k \in \mathbb{N}^+$, *if* $|\mathcal{X}| = m$ *is finite, then there exists an approximate posterior* $\widetilde{P}_{X^n}(x|y = y^*)$ *satisfies that for any* $s \in \{1, \cdots, m\}$,

$$\mathbb{E}_{X^n} \left\{ \widetilde{P}_{X^n}(x = u_s|y = y^*) \right\} - P_{X|Y}(u_s|y^*) = \mathcal{O}_{k,m,G}(n^{-k}),$$
$$\mathrm{Var}_{X^n} \left\{ \widetilde{P}_{X^n}(x = u_s|y = y^*) \right\} = \mathcal{O}_{k,m,G}(n^{-1}).$$

The proposed sampling scheme in Algorithm 1 generates $k$ samples and a linear combination of whose distributions approximates the posterior. In applications where it is desired to still generate one sample (rather than using a linear combination), we may consider a rejection sampling algorithm based on Theorem 3 to sample from $\widetilde{P}_{X^n}(x|y = y^*)$. Let $\mathbf{T} = (T_1, \cdots, T_m)^\top$ where $T_j = \sum_{i=1}^n \mathbb{1}_{X_i = u_j}$ for $j = 1, \cdots, m$. Then $(g_1(\mathbf{T}/n), \cdots, g_m(\mathbf{T}/n))^\top$ is the posterior probability vector associated with the plug-in posterior $\widehat{P}_{X^n}(x|y = y^*)$ and $(D_{n,k}(g_1)(\mathbf{T}/n), \cdots, D_{n,k}(g_m)(\mathbf{T}/n))^\top$ is the posterior probability vector associated with the debiased posterior $\widetilde{P}_{X^n}(x|y = y^*)$. The rejection sampling is described in Algorithm 2.

---

**Algorithm 2** Rejection Sampling for Debiased Posterior $\widetilde{P}_{X^n}(x \mid y = y^*)$

---

**Input:** Plug-in posterior $\widehat{P}_{X^n}(x \mid y = y^*)$, debiased posterior $\widetilde{P}_{X^n}(x \mid y = y^*)$, large enough constant $M > 0$

**Output:** Sample from the debiased posterior $\widetilde{P}_{X^n}(x \mid y = y^*)$

1: **repeat**
2:     Sample $x' \sim \widehat{P}_{X^n}(x \mid y = y^*)$
3:     Sample $u \sim \text{Uniform}(0, M)$
4: **until** $u < \dfrac{\widetilde{P}_{X^n}(x' \mid y = y^*)}{\widehat{P}_{X^n}(x' \mid y = y^*)}$
5: **return** $x'$

---

In Algorithm 2,

$$M = \max_{x \in \mathcal{X}} \frac{\widetilde{P}_{X^n}(x|y = y^*)}{\widehat{P}_{X^n}(x|y = y^*)} = \max_j \left\{ \frac{D_{n,k}(g_j)(\mathbf{T}/n)}{g_j(\mathbf{T}/n)} \right\}$$

is the upper bound of the ratio of the debiased posterior to the plug-in posterior.

## 5 Experiments

In this section, we provide numerical experiments to illustrate the debiasing framework for posterior approximation under the binary prior case and the Gaussian mixture prior case.

**Binary prior case**. Suppose that $\mathcal{X} = \{0, 1\}$ and $X \sim \text{Bern}(q)$ for some unknown prior $q \in (0, 1)$. Let $\alpha = \alpha(y^*) := P_{Y|X}(y^*|1)/P_{Y|X}(y^*|0)$ be the likelihood ratio. Then the posterior distribution is give by $X|Y \sim \text{Bern}(\alpha q/(\alpha q + 1 - q))$. We estimate $g(q) := \alpha q/(\alpha q + 1 - q)$ based on the observations of $X^n$ and $y^*$.

Proposition 1 provides a debiased approximation as a special case of Theorem 2 when $|\mathcal{X}| = 2$.

**Proposition 1.** *Let* $T = \sum_{i=1}^n X_i$. *For* $k = 1, 2, 3, 4$, *we have*

$$\mathbb{E}_{X^n} \{D_{n,k}g(T/n)\} - g(q) = \mathcal{O}(n^{-k}),$$

*where* $D_{n,k} = \sum_{j=0}^{k-1} \binom{k}{j+1}(-1)^j B_n^j$ *and* $B_n(g)(x) = \sum_{k=0}^n g(\frac{k}{n})\binom{n}{k}x^k(1-x)^{n-k}$ *is the Bernstein polynomial approximation of g.*

In the proof of Theorem 2, we notice that for any $k \in \mathbb{N}^+$, $\mathbb{E}_{X^n} \{D_{n,k}g(T/n)\} = C_{n,k}g(q)$, which allows Proposition 1 to be verified in closed form. To validate this result numerically, we consider two parameter settings: in the first experiment we set $q = 0.4$, $y^* = 2$, and $Y|X \sim \mathcal{N}(X, 1)$, while in the second we set $q = 3/11$, $y^* = 1$, and $Y|X \sim \mathcal{N}(X, 1/4)$.

For both settings, we examine the convergence rate of the debiased estimators $D_{n,k}g(T/n)$ for $k = 1, 2, 3, 4$. The results are shown in log-log plots in Figure 1, where the vertical axis represents the logarithm of the absolute error and the horizontal axis represents the logarithm of the sample size $n$. Reference lines with slopes corresponding to $n^{-1}, n^{-2}, n^{-3}$, and $n^{-4}$ are included for comparison.

**Gaussian mixture prior case**. Suppose that $X \sim \frac{1}{2}\mathcal{N}(0, 1) + \frac{1}{2}\mathcal{N}(1, 1)$ and $Y = X + \xi$ where $\xi \sim \mathcal{N}(0, 1/16)$. Additionally, let $y^* = 0.8$ and $A = \{x : x \geq 0.5\}$. In this case, we validate the theoretical convergence rate

$$|\mathbb{E}_{X^n} \{D_{n,k}f(\hat{\pi})(A)\} - f(\pi)(A)| = \mathcal{O}(n^{-k}).$$

Since $\mathbb{E}_{X^n}\{D_{n,k}f(\hat{\pi})(A)\}$ does not have a closed-form expression, we approximate it using Monte Carlo simulation. To ensure that the Monte Carlo error is negligible compared to the bias $\mathcal{O}(n^{-k})$, we select the number of Monte Carlo samples $N$ such that $N \gg n^{2k-1}$. In practice, we run simulations for $k = 1$ and $k = 2$ and set $N = n^3$ for $k = 1$ and $N = n^4$ for $k = 2$.

The results are shown in Figure 2. The figure presents log-log plots where the vertical axis represents the logarithm of the absolute error or of the variance and the horizontal axis represents the logarithm of the sample size $n$. For both $k = 1$ and $k = 2$, the observed convergence rates align closely with the theoretical predictions.

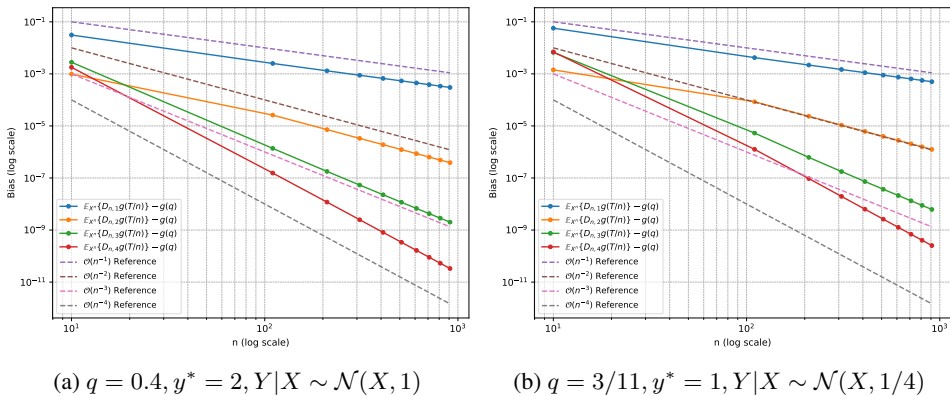

(a) $q = 0.4, y^* = 2, Y|X \sim \mathcal{N}(X, 1)$      (b) $q = 3/11, y^* = 1, Y|X \sim \mathcal{N}(X, 1/4)$

Figure 1: Convergence of plug-in and debiased estimators in the binary prior case. The plot compares the approximation error of $D_{n,k}g(T/n)$ ($k = 1, 2, 3, 4$) against $n$. Reference lines with slopes corresponding to $n^{-1}, n^{-2}, n^{-3}$, and $n^{-4}$ are included to highlight the convergence rates.

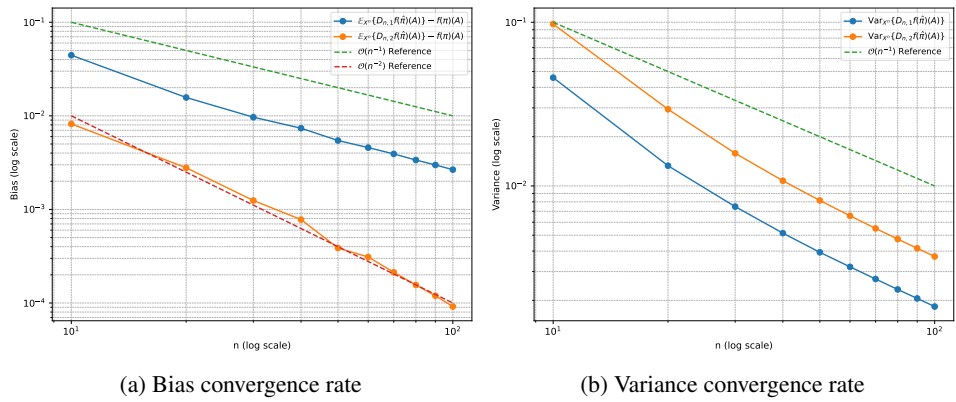

(a) Bias convergence rate        (b) Variance convergence rate

Figure 2: Convergence of debiased estimators in the Gaussian mixture prior case with $X \sim \frac{1}{2}\mathcal{N}(0, 1) + \frac{1}{2}\mathcal{N}(1, 1)$, $Y = X + \xi$, $\xi \sim \mathcal{N}(0, 1/16)$, $y^* = 0.8$, and $A = \{x : x \geq 0.5\}$. (a) shows the bias decay of $D_{n,k}f(\hat{\pi})(A)$ for $k = 1, 2$, with reference lines of slopes corresponding to $n^{-1}$ and $n^{-2}$ included for comparison. (b) shows the corresponding variance decay, alongside a reference slope corresponding to $n^{-1}$.

## 6 Conclusion

We introduced a general framework for constructing a debiased posterior approximation through observed samples $D$ and the known likelihood $P_{Y|X}$ when the prior distribution is unknown. Here, a naive strategy that directly plugs the empirical distribution into the Bayes formula or a generative model has a bias, because the likelihood is nonconstant, inducing a distribution shift, and the map from the prior to posterior is nonlinear. It can be shown that the plug-in approach generates $\hat{X}$ with bias $\|\mathbb{E}_D(P_{\hat{X}|Y=y^*,D}) - P_{X|Y=y^*}\|_{\text{TV}} = \mathcal{O}(n^{-1})$ and variance $\text{Var}_D(P_{\hat{X}|Y=y^*,D}) = \mathcal{O}(n^{-1})$. In contrast, our proposed debiasing framework achieves arbitrarily high-order bias rate of $\mathcal{O}(n^{-k})$ for any integer $k$, while maintaining the order of magnitude of the variance. Our framework is black-box in the sense that we only need to resample the training data and feed it into a given black-box conditional generative model. In particular, we provide a rigorous proof for the Bayes-optimal sampler $f$ under the continuous prior setting and for a broad class of samplers $f$ with a continuous $2k$-th derivative, as specified in Assumption 2, under the discrete prior setting. We expect that the proposed debiasing framework could work for general $f$ and will support future developments in bias-corrected posterior estimation and conditional sampling.

## Acknowledgments

This research was supported in part by NSF Grant DMS-2515510.

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

# A Proof of Theorem 1

*Proof of Theorem 1.* We begin by introducing notations that facilitates the analysis. Define

$$\mu = \int_{\mathcal{X}} \ell(x)\pi(dx), \quad \mu_A = \int_A \ell(x)\pi(dx).$$

Let $\hat{\pi}^{(0)} = \pi, \hat{\pi}^{(1)} = \hat{\pi}$ and set $(X_1^{(0)}, \ldots, X_n^{(0)}) \equiv (X_1, \ldots, X_n)$. For $j = 1, \ldots, k$, we define $\hat{\pi}^{(j)}$ as the empirical measure of $n$ i.i.d. samples $(X_1^{(j-1)}, \ldots, X_n^{(j-1)})$ drawn from the measure $\hat{\pi}^{(j-1)}$. Furthermore, for each $j = 0, \ldots, k$, define

$$e_n^{(j)} = n^{-1} \sum_{i=1}^n \left\{ \ell(X_i^{(j)}) - \mu \right\}, \quad \mu_A^{(j)} = n^{-1} \sum_{i=1}^n \ell(X_i^{(j)}) \delta_{X_i^{(j)}}(A).$$

Let

$$C_{n,k} = \sum_{j=1}^k \binom{k}{j} (-1)^{j-1} B_n^j,$$

so that it suffices to show that

$$C_{n,k} f(\pi)(A) - f(\pi)(A) = \mathcal{O}_{L_1, L_2, k}(n^{-k}) \tag{9}$$

since $B_n D_{n,k} = C_{n,k}$.

The Radon-Nikodym derivative of $f(\pi)$ with respect to $\pi$ is

$$\frac{df(\pi)}{d\pi}(x) = \frac{\ell(x)}{\displaystyle\int_{\mathcal{X}} \ell(x)\pi(dx)}.$$

For the empirical measure $\hat{\pi}$, the corresponding Radon-Nikodym derivative of $f(\hat{\pi})$ with respect to $\hat{\pi}$ takes the form

$$\frac{df(\hat{\pi})}{d\hat{\pi}}(x) = \begin{cases} \dfrac{\ell(x)}{\displaystyle\int_{\mathcal{X}} \ell(x)\hat{\pi}(dx)}, & \text{if } x \in \left\{ X_1^{(0)}, \ldots, X_n^{(0)} \right\}, \\ 0, & \text{otherwise}, \end{cases}$$

$$= \begin{cases} \dfrac{\ell(x)}{n^{-1} \sum_{i=1}^n \ell(X_i^{(0)})}, & \text{if } x \in \left\{ X_1^{(0)}, \ldots, X_n^{(0)} \right\}, \\ 0, & \text{otherwise}. \end{cases}$$

Consequently,

$$\begin{aligned}
B_n f(\pi)(A) &= \mathbb{E}\left\{ f(\hat{\pi})(A) \right\} \\
&= \mathbb{E}\left\{ \int_A \frac{df(\hat{\pi})}{d\hat{\pi}}(x)\hat{\pi}(dx) \right\} \\
&= \mathbb{E}\left\{ \int_A \frac{\ell(x)}{n^{-1}\sum_{i=1}^n l(X_i^{(0)})}\hat{\pi}(dx) \right\} \\
&= \mathbb{E}\left\{ \frac{n^{-1}\sum_{i=1}^n \ell(X_i^{(0)})\delta_{X_i^{(0)}}(A)}{n^{-1}\sum_{i=1}^n \ell(X_i^{(0)})} \right\}.
\end{aligned}$$

Moreover, by the definition of $B_n$ and iterated conditioning, we have $\mathbb{E}\left\{ f(\hat{\pi}^{(j)})(A) \right\} = \mathbb{E}\left[\mathbb{E}\left\{ f(\hat{\pi}^{(j)})(A)|\hat{\pi}^{(j-1)}\right\}\right] = \mathbb{E}\left\{ B_n f(\hat{\pi}^{(j-1)})(A)\right\} = \cdots = \mathbb{E}\left\{ B_n^{j-1} f(\hat{\pi}^{(1)})(A)\right\} = B_n^j f(\pi)(A)$.

By the same logic, for any $j = 2, \ldots, k$, we have

$$\mathbb{E}\left\{f(\hat{\pi}^{(j)})(A)\right\} = \mathbb{E}\left\{\frac{n^{-1}\sum_{i=1}^{n}\ell(X_i^{(j-1)})\delta_{X_i^{(j-1)}}(A)}{n^{-1}\sum_{i=1}^{n}\ell(X_i^{(j-1)})}\right\}.$$

Thus,

$$\begin{aligned}
C_{n,k}f(\pi)(A) &= \sum_{j=1}^{k}\binom{k}{j}(-1)^{j-1}B_n^j f(\pi)(A) \\
&= \sum_{j=1}^{k}\binom{k}{j}(-1)^{j-1}\mathbb{E}\left\{f(\hat{\pi}^{(j)})(A)\right\} \\
&= \sum_{j=1}^{k}\binom{k}{j}(-1)^{j-1}\mathbb{E}\left\{\frac{n^{-1}\sum_{i=1}^{n}\ell(X_i^{(j-1)})\delta_{X_i^{(j-1)}}(A)}{n^{-1}\sum_{i=1}^{n}\ell(X_i^{(j-1)})}\right\} \\
&= \sum_{j=1}^{k}\binom{k}{j}(-1)^{j-1}\mathbb{E}\left(\frac{\mu_A^{(j-1)}}{e_n^{(j-1)}+\mu}\right).
\end{aligned}$$

Then (9) holds if

$$\sum_{j=1}^{k}\binom{k}{j}(-1)^{j-1}\mathbb{E}\left(\frac{\mu_A^{(j-1)}}{e_n^{(j-1)}+\mu}\right) = \frac{\mu_A}{\mu} + \mathcal{O}_{L_1,L_2,k}(n^{-k}). \tag{10}$$

Now we show that (10) holds. By using the Taylor expansion of $1/(e_n^{(j-1)}+\mu)$, we have

$$\frac{1}{e_n^{(j-1)}+\mu} = \frac{1}{\mu} + \sum_{r=1}^{2k-1}\frac{(-1)^r}{\mu^{r+1}}(e_n^{(j-1)})^r + \frac{(e_n^{(j-1)})^{2k}}{\xi^{2k+1}},$$

where $\xi$ lies between $e_n^{(j-1)}+\mu$ and $\mu$.

Since $\min\{e_n^{(j-1)}+\mu, \mu\} \geq L_1$, we have $1/\xi^{2k+1} \leq L_1^{-2k-1}$. Additionally, $L_1 \leq l(X_i^{(j-1)}) \leq L_2$ and Hoeffding's inequality implies that

$$\mathbb{P}(|ne_n^{(j-1)}| > t) \leq 2\exp\left\{-\frac{2t^2}{n(L_2-L_1)^2}\right\}$$

for all $t > 0$, which is equivalent to

$$\mathbb{P}(|e_n^{(j-1)}| > t) \leq 2\exp\left\{-\frac{2nt^2}{(L_2-L_1)^2}\right\}.$$

Then

$$\begin{aligned}
\mathbb{E}\left(|e_n^{(j-1)}|^{2k}\right) &= \int_0^\infty \mathbb{P}\left(\left|e_n^{(j-1)}\right|^{2k} > t\right)dt \\
&= \int_0^\infty \mathbb{P}\left(\left|e_n^{(j-1)}\right| > t^{1/2k}\right)dt \\
&\leq \int_0^\infty 2ku^{k-1}\exp\left\{-\frac{2nu}{(L_2-L_1)^2}\right\}du \\
&= 2kn^{-k}\int_0^\infty \exp\left\{-\frac{2v}{(L_2-L_1)^2}\right\}v^{k-1}dv \\
&= \mathcal{O}_{L_1,L_2,k}(n^{-k}).
\end{aligned}$$

Therefore, we have

$$\mathbb{E}\left(\frac{\mu_A^{(j-1)}}{e_n^{(j-1)}+\mu}\right) = \frac{\mu_A}{\mu} + \mathbb{E}\left\{\sum_{r=1}^{2k-1}\frac{(-1)^r}{\mu^{r+1}}\mu_A^{(j-1)}(e_n^{(j-1)})^r\right\} + \mathcal{O}_{L_1,L_2,k}(n^{-k}),$$

which implies that the L.H.S. of (10) can be written as

$$\frac{\mu_A}{\mu} + \sum_{r=1}^{2k-1}\frac{(-1)^r}{\mu^{r+1}}\left[\sum_{j=1}^{k}\binom{k}{j}(-1)^{j-1}\mathbb{E}\left\{\mu_A^{(j-1)}(e_n^{(j-1)})^r\right\}\right] + \mathcal{O}_{L_1,L_2,k}(n^{-k}).$$

Thus to prove the bound on the bias, it remains to show that for any given $k$ and $1 \le r \le 2k-1$,

$$B_{k,r} := \sum_{j=1}^{k}\binom{k}{j}(-1)^{j-1}\mathbb{E}\left\{\mu_A^{(j-1)}(e_n^{(j-1)})^r\right\} = \mathcal{O}_{L_1,L_2,k}(n^{-k}).$$

Define a new operator $B : h \mapsto \mathbb{E}\{h(\hat{\pi})\}$ for any $h : \Delta_{\mathcal{X}} \to \mathbb{R}$ and let

$$h_s(\pi) = \left\{\int_A \ell(x)\pi(dx)\right\}\left\{\int \ell(x)\pi(dx)\right\}^s.$$

Since $B^j h_s(\pi) = \mathbb{E}\left\{h_s(\hat{\pi}^{(j)})\right\}$, we have

$$B_{k,r} = \sum_{j=1}^{k}\binom{k}{j}(-1)^{j-1}\sum_{s=0}^{r}\binom{r}{s}B^j h_s(\pi)(-1)^{(r-s)}\mu^{r-s}$$

$$= \sum_{s=0}^{r}\binom{r}{s}(-1)^{r-s}\mu^{r-s}\sum_{j=1}^{k}\binom{k}{j}(-1)^{j-1}B^j h_s(\pi).$$

We claim that $B_{k,r} = \mathcal{O}_{L_1,L_2,k}(n^{-k})$ holds if for any $0 \le s \le r \le 2k-1$,

$$(I-B)^k h_s(\pi) = \mathcal{O}_{L_1,L_2,s}(n^{-k}). \tag{11}$$

Indeed, $(I-B)^k h_s(\pi) = \mathcal{O}_{L_1,L_2,s}(n^{-k})$ is equivalent to $\sum_{j=1}^{k}\binom{k}{j}(-1)^{j-1}B^j h_s(\pi) = h_s(\pi) + \mathcal{O}_{L_1,L_2,s}(n^{-k})$. Therefore (11) implies

$$B_{k,r} = \sum_{s=0}^{r}\binom{r}{s}(-1)^{r-s}\mu^{r-s}\left\{h_s(\pi) + \mathcal{O}_{L_1,L_2,s}(n^{-k})\right\}$$

$$= \sum_{s=0}^{r}\left\{\binom{r}{s}(-1)^{r-s}\mu_A\mu^r + \mathcal{O}_{L_1,L_2,s}(n^{-k})\right\}$$

$$= \mathcal{O}_{L_1,L_2,k}(n^{-k}).$$

Now, to prove the bound on the bias we only need to show that (11) holds. For any $k \in \mathbb{N}$ and $s \in \mathbb{N}^+$, let

$$\mathfrak{J}_s := \left\{(\mathbf{a},\mathbf{s},v): \mathbf{a} = (a_1,a_2,\ldots), \mathbf{s} = (s_1,s_2,\ldots), a_i, s_i \in \mathbb{N}^+, v \in \mathbb{N}, a_1 > a_2 > \cdots \ge 1, \sum_i a_i s_i + v = s\right\}$$

and

$$\mathcal{A}_s^k := \left\{\sum_{(\mathbf{a},\mathbf{s},v)\in\mathfrak{J}_s}\alpha_{\mathbf{a},\mathbf{s},v}\left\{\int_A \ell^v(x)\pi(dx)\right\}\prod_i\left\{\int \ell^{a_i}(x)\,\pi(dx)\right\}^{s_i} : |\alpha_{\mathbf{a},\mathbf{s},v}| \le C_k(s)n^{-k}\right\},$$

where $C_0(s), C_1(s), \ldots$ are constants from Lemma 2. Since $h_s(\pi) \in \mathcal{A}_{s+1}^0$, Lemma 2 implies that $(I-B)^k h_s(\pi) \in \mathcal{A}_{s+1}^k$. Therefore, $(I-B)^k h_s(\pi) = \mathcal{O}_{L_1,L_2,s}(n^{-k})$, finishing the proof for the bias bound.

Finally, to prove the bound on the variance, consider the function $F(x, y) = x/y$. By construction,

$$f(\hat{\pi}^{(j)})(A) = F\left(\mu_A^{(j-1)}, e_n^{(j-1)} + \mu\right).$$

Applying the Taylor expansion of $F(x, y)$ yields

$$f(\hat{\pi}^{(j)})(A) = f(\pi)(A) + \frac{1}{\mu_A}(\mu_A^{(j-1)} - \mu_A) - \frac{\mu_A}{\mu^2}e_n^{(j-1)} - \frac{1}{\xi_y^2}(\mu_A^{(j-1)} - \mu_A)e_n^{(j-1)} + \frac{\xi_x}{\xi_y^3}\left(e_n^{(j-1)}\right)^2,$$

for some $\xi_x$ lying between $\mu_A$ and $\mu_A^{(j-1)}$, and $\xi_y$ lying between $\mu$ and $e_n^{(j-1)} + \mu$. Since $L_1 \leq \mu_A, \mu_A^{(j-1)}, \mu, e_n^{(j-1)} + \mu \leq L_2$ implies that $|1/\xi_y^2|$ and $|\xi_x/\xi_y^3|$ are bounded by some constant depending on $L_1$ and $L_2$.

Moreover, since

$$\left|\mathbb{E}\left\{(\mu_A^{(j-1)} - \mu_A)e_n^{(j-1)}\right\}\right| = \left|\mathrm{Cov}\left(\mu_A^{(j-1)}, e_n^{(j-1)} + \mu\right)\right|$$

$$\leq \left\{\mathrm{Var}\left(\mu_A^{(j-1)}\right)\right\}^{1/2}\left\{\mathrm{Var}\left(e_n^{(j-1)} + \mu\right)\right\}^{1/2}$$

$$= \left[\mathrm{Var}\left\{n^{-1}\sum_{i=1}^{n}\ell(X_i^{(j)})\delta_{X_i^{(j-1)}}(A)\right\}\right]^{1/2}\left[\mathrm{Var}\left\{n^{-1}\sum_{i=1}^{n}\ell(X_i^{(j-1)})\right\}\right]^{1/2}$$

$$= \left[\frac{1}{n}\mathrm{Var}\left\{\ell(X_i^{(j)})\delta_{X_i^{(j-1)}}(A)\right\}\right]^{1/2}\left[\frac{1}{n}\mathrm{Var}\left\{\ell(X_i^{(j-1)})\right\}\right]^{1/2}$$

$$= \mathcal{O}_{L_1,L_2}(n^{-1}),$$

and

$$\mathbb{E}\left(e_n^{(j-1)}\right)^2 = \frac{1}{n}\mathrm{Var}\left\{\ell(X_i^{(j-1)})\right\} = \mathcal{O}_{L_1,L_2}(n^{-1}).$$

Combining these bounds with the Taylor expansion, we conclude that for any $j \geq 1$,

$$B_n^j f(\pi)(A) = \mathbb{E}\left\{f(\hat{\pi}^{(j)})(A)\right\} = f(\pi)(A) + \mathcal{O}_{L_1,L_2}(n^{-1}).$$

By the same logic, we also have $B_n\left\{f(\pi)(A)\right\}^2 = \left\{f(\pi)(A)\right\}^2 + \mathcal{O}_{L_1,L_2}(n^{-1})$.

Therefore,

$$D_{n,k}f(\pi)(A) = \sum_{j=0}^{k-1}\binom{k}{j+1}(-1)^j B_n^j f(\pi)(A)$$

$$= \sum_{j=0}^{k-1}\binom{k}{j+1}(-1)^j\left\{f(\pi)(A) + \mathcal{O}_{L_1,L_2}(n^{-1})\right\}$$

$$= f(\pi)(A) + \mathcal{O}_{L_1,L_2,k}(n^{-1}),$$

and

$$\mathrm{Var}_{X^n}\left\{D_{n,k}f(\hat{\pi})(A)\right\} = \mathbb{E}\left[\left\{D_{n,k}f(\hat{\pi})(A)\right\}^2\right] - \left[\mathbb{E}\left\{D_{n,k}f(\hat{\pi})(A)\right\}\right]^2$$

$$= B_n\left\{D_{n,k}f(\pi)(A)\right\}^2 - \left\{f(\pi)(A) + \mathcal{O}_{L_1,L_2,k}(n^{-k})\right\}^2$$

$$= B_n\left\{f(\pi)(A) + \mathcal{O}_{L_1,L_2,k}(n^{-1})\right\}^2 - \left\{f(\pi)(A) + \mathcal{O}_{L_1,L_2,k}(n^{-k})\right\}^2$$

$$= B_n\left\{f(\pi)(A)\right\}^2 + \mathcal{O}_{L_1,L_2,k}(n^{-1}) - \left\{f(\pi)(A)\right\}^2$$

$$= \mathcal{O}_{L_1,L_2,k}(n^{-1}).$$

$\square$

**Lemma 2.** *There exist constants $C_0(s)$, $C_1(s)$, $C_2(s), \ldots$, such that the following holds.*
*For any $k \in \mathbb{N}$ and $s, n \in \mathbb{N}^+$, let*

$$\mathfrak{J}_s := \left\{ (\mathbf{a}, \mathbf{s}, v) \colon \mathbf{a} = (a_1, a_2, \ldots), \mathbf{s} = (s_1, s_2, \ldots), a_i, s_i \in \mathbb{N}^+, v \in \mathbb{N}, a_1 > a_2 > \cdots \geq 1, \sum_i a_i s_i + v = s \right\}$$

*and*

$$\mathcal{A}_s^k := \left\{ \sum_{(\mathbf{a}, \mathbf{s}, v) \in \mathfrak{J}_s} \alpha_{\mathbf{a}, \mathbf{s}, v} \left\{ \int_A \ell^v(x) \pi(dx) \right\} \prod_i \left\{ \int \ell^{a_i}(x) \, \pi(dx) \right\}^{s_i} \colon |\alpha_{\mathbf{a}, \mathbf{s}, v}| \leq C_k(s) n^{-k} \right\}.$$

*If $h(\pi) \in \mathcal{A}_s^0$, then for any $k \in \mathbb{N}$, we have*

$$(I - B)^k h(\pi) \in \mathcal{A}_s^k, \tag{12}$$

*where $B$ is an operator defined as $Bh(\pi) = \mathbb{E}\{h(\hat{\pi})\}$ where $\hat{\pi}$ is the empirical distribution of $X_1, X_2, \ldots, X_n \overset{\text{i.i.d.}}{\sim} \pi$.*

*Proof of Lemma 2.* We begin by proving that $(I - B)h(\pi) \in \mathcal{A}_s^1$. Since $h(\pi) \in \mathcal{A}_s^0$, let

$$h(\pi) = \sum_{(\mathbf{a}, \mathbf{s}, v) \in \mathfrak{J}_s} \alpha_{\mathbf{a}, \mathbf{s}, v} \left\{ \int_A \ell^v(x) \pi(dx) \right\} \prod_i \left\{ \int \ell^{a_i}(x) \, \pi(dx) \right\}^{s_i}.$$

Note that $|\mathfrak{J}_s|$ does not depend on $n$ and $|\alpha_{\mathbf{a}, \mathbf{s}, v}| \leq C_0(s)$. It suffices to verify that each individual term in the sum satisfies

$$(I - B) \left[ \left\{ \int_A \ell^v(x) \pi(dx) \right\} \prod_i \left\{ \int \ell^{a_i}(x) \, \pi(dx) \right\}^{s_i} \right] \in \mathcal{A}_s^1.$$

Without loss of generality, let $\mathbf{a} = (a_1, \ldots, a_p)$ and $\mathbf{s} = (s_1, \ldots, s_p)$, $s' = \sum_{i=1}^p s_i$. Then we have $\sum_i^p a_i s_i + v = s$ and

$$B \left[ \left\{ \int_A \ell^v(x) \pi(dx) \right\} \prod_{i=1}^p \left\{ \int \ell^{a_i}(x) \, \pi(dx) \right\}^{s_i} \right]$$

$$= \mathbb{E} \left[ \left\{ \int_A \ell^v(x) \hat{\pi}(dx) \right\} \prod_{i=1}^p \left\{ \int \ell^{a_i}(x) \, \hat{\pi}(dx) \right\}^{s_i} \right]$$

$$= \frac{1}{n^{s'+1}} \mathbb{E} \left[ \left\{ \sum_{j=1}^n \ell^v(X_j) \delta_{X_j}(A) \right\} \prod_{i=1}^p \left\{ \sum_{j=1}^n \ell^{a_i}(X_j) \right\}^{s_i} \right].$$

For the term $\prod_{i=1}^p \left\{ \sum_{j=1}^n \ell^{a_i}(X_j) \right\}^{s_i}$, let $m_j^{(i)}$ denote the times $X_j$ appears with powers $a_i$, then we have $\sum_{j=1}^n m_j^{(i)} = s_i$ for $1 \leq i \leq p$. Define

$$\mathcal{I}_\mathbf{s} = \left\{ \mathbf{m} = \left( m_j^{(i)} \right)_{j \in [n], i \in [p]} \colon \sum_{j=1}^n m_j^{(i)} = s_i \text{ for all } i \in [p] \right\}.$$

Therefore,

$$\left\{ \sum_{j=1}^n \ell^v(X_j) \delta_{X_j}(A) \right\} \prod_{i=1}^p \left\{ \sum_{j=1}^n \ell^{a_i}(X_j) \right\}^{s_i} = \left\{ \sum_{j=1}^n \ell^v(X_j) \delta_{X_j}(A) \right\} \sum_{\mathbf{m} \in \mathcal{I}_\mathbf{s}} c_{\mathbf{s}, \mathbf{m}} \prod_{j=1}^n \ell^{\sum_{i=1}^p a_i m_j^{(i)}}(X_j), \tag{13}$$

where

$$c_{\mathbf{s}, \mathbf{m}} = \prod_{i=1}^p \frac{s_i!}{\prod_{j=1}^n m_j^{(i)}!}.$$

Note that $c_{\mathbf{s},\mathbf{m}}$ does not depend on $n$. Now we expand R.H.S. of (13) based on the number of distinct variables $X_j$ appear, i.e., $\sum_{j=1}^n \mathbb{1}_{\sum_{i=1}^p a_i m_j^{(i)} > 0}$, which is equal to $\sum_{j=1}^n \mathbb{1}_{\sum_{i=1}^p m_j^{(i)} > 0}$. Define

$$\mathcal{J}_{\mathbf{m}} = \left\{ j \in [n]\colon \sum_{i=1}^p m_j^{(i)} > 0 \right\},$$

then we have $1 \le |\mathcal{J}_{\mathbf{m}}| \le s'$.

Hence,

$$\mathbb{E}\left[ \left\{ \sum_{j=1}^n \ell^v(X_j)\delta_{X_j}(A) \right\} \prod_{i=1}^p \left\{ \sum_{j=1}^n \ell^{a_i}(X_j) \right\}^{s_i} \right]$$

$$= \mathbb{E}\left[ \left\{ \sum_{j=1}^n \ell^v(X_j)\delta_{X_j}(A) \right\} \sum_{m=1}^{s'} \sum_{\substack{\mathbf{m}\in\mathcal{I}_{\mathbf{s}} \\ |\mathcal{J}_{\mathbf{m}}|=m}} c_{\mathbf{s},\mathbf{m}} \prod_{j=1}^n \ell^{\sum_{i=1}^p a_i m_j^{(i)}}(X_j) \right]$$

$$= \mathbb{E}\left\{ \sum_{m=1}^{s'} \sum_{\substack{\mathbf{m}\in\mathcal{I}_{\mathbf{s}} \\ |\mathcal{J}_{\mathbf{m}}|=m}} c_{\mathbf{s},\mathbf{m}} \sum_{t=1}^n \ell^v(X_t)\delta_{X_t}(A) \prod_{j=1}^n \ell^{\sum_{i=1}^p a_i m_j^{(i)}}(X_j) \right\}$$

$$= n(n-1)\cdots(n-s')c_{\mathbf{s},\mathbf{m}^*} \left\{ \int_A \ell^v(x)\pi(dx) \right\} \prod_{i=1}^p \left[\mathbb{E}\left\{ \ell^{a_i}(X) \right\}\right]^{s_i}$$

$$+ \mathbb{E}\left\{ \sum_{\substack{\mathbf{m}\in\mathcal{I}_{\mathbf{s}} \\ |\mathcal{J}_{\mathbf{m}}|=s'}} c_{\mathbf{s},\mathbf{m}} \sum_{t\in\mathcal{J}_{\mathbf{m}}} \ell^v(X_t)\delta_{X_t}(A) \prod_{j=1}^n \ell^{\sum_{i=1}^p a_i m_j^{(i)}}(X_j) \right\}$$

$$+ \mathbb{E}\left\{ \sum_{m=1}^{s'-1} \sum_{\substack{\mathbf{m}\in\mathcal{I}_{\mathbf{s}} \\ |\mathcal{J}_{\mathbf{m}}|=m}} c_{\mathbf{s},\mathbf{m}} \sum_{t=1}^n \ell^v(X_t)\delta_{X_t}(A) \prod_{j=1}^n \ell^{\sum_{i=1}^p a_i m_j^{(i)}}(X_j) \right\}, \quad (14)$$

where $c_{\mathbf{s},\mathbf{m}^*} = \prod_{i=1}^p s_i!$.

The three terms in (14) are interpreted as follows: we can expand $\{\sum_{t=1}^n \ell^v(X_t)\delta_{X_t}(A)\} \prod_{i=1}^p \left\{\sum_{j=1}^n \ell^{a_i}(X_j)\right\}^{s_i}$ as the sum of many product terms of the form $\ell^v(X_t) \prod_{i=1}^p \prod_{l=1}^{s_i} \ell^{a_i}(X_{j_{i,l}})$. The first term in (14) corresponds to the partial sum of terms in which all of $X_t, (X_{j_{i,l}})_{i,l}$ are distinct. The second term in (14) corresponds to the partial sum of terms in which $X_t$ is identical to one of $(X_{j_{i,l}})_{i,l}$ while the latter are distinct. The third term corresponds to the partial sum of terms in which at least two of $(X_{j_{i,l}})_{i,l}$ are identical. The last two term in (14) are at least $\mathcal{O}(n^{-1})$ factor smaller than the first (due to fewer terms involved in the sum because of the constraint of having identical terms), while the first term will cancel with $I \cdot h(\pi)$ when applying $I - B$ to $h$.

Let $\mathcal{P}(b_1, \ldots, b_m)$ denote the set of all distinct permutations of the vector consisting of $m$ non-zero elements $b_1, \ldots, b_m$ and $n - m$ zeros. Note that even the values of $b_i$ may be the same, we still treat the $b_i$s are distinguishable. Then since 0s are identical, we have $|\mathcal{P}(b_1, \ldots, b_m)| = n(n-1)\cdots(n-m+1) = \mathcal{O}(n^m)$. Additionally, for any $\mathbf{a}$ and $\mathbf{m}$, we define

$$\Psi(\mathbf{a},\mathbf{m}) = \left( \sum_{i=1}^p a_i m_1^{(i)}, \ldots, \sum_{i=1}^p a_i m_n^{(i)} \right).$$

Now we can write (14) as

$$n(n-1)\cdots(n-s')c_{\mathbf{s},\mathbf{m}^*} \left\{ \int_A \ell^v(x)\pi(dx) \right\} \prod_{i=1}^p \left[\mathbb{E}\{\ell^{a_i}(X)\}\right]^{s_i}$$

$$+ \sum_{b_k : \sum_{k=1}^{s'} b_k = s - v} \ \sum_{\mathbf{m} : \Psi(\mathbf{a},\mathbf{m}) \in \mathcal{P}(b_1,\ldots,b_{s'})} c_{\mathbf{s},\mathbf{m}} \sum_{t=1}^{m} \left[ \prod_{i \neq t} \mathbb{E}\{\ell^{b_i}(X)\} \right] \int_A \ell^{b_t + v}(x) \pi(dx)$$

$$+ \sum_{m=1}^{s'-1} \ \sum_{b_k : \sum_{k=1}^{m} b_k = s - v} \ \sum_{\mathbf{m} : \Psi(\mathbf{a},\mathbf{m}) \in \mathcal{P}(b_1,\ldots,b_m)} c_{\mathbf{s},\mathbf{m}} \mathbb{E} \left\{ \sum_{t=1}^{n} \ell^v(X_t) \delta_{X_t}(A) \prod_{i=1}^{m} \ell^{b_i}(X_i) \right\}$$

$$= n(n-1) \cdots (n - s') c_{\mathbf{s},\mathbf{m}^*} \left\{ \int_A \ell^v(x) \pi(dx) \right\} \prod_{i=1}^{p} [\mathbb{E}\{\ell^{a_i}(X)\}]^{s_i}$$

$$+ \sum_{b_k : \sum_{k=1}^{s'} b_k = s - v} \mathcal{O}(n^{s'}) c_{\mathbf{s},\mathbf{m}} \sum_{t=1}^{m} \left[ \prod_{i \neq t} \mathbb{E}\{\ell^{b_i}(X)\} \right] \int_A \ell^{b_t + v}(x) \pi(dx)$$

$$+ \sum_{m=1}^{s'-1} \ \sum_{b_k : \sum_{k=1}^{m} b_k = s - v} \mathcal{O}(n^m) c_{\mathbf{s},\mathbf{m}} \mathbb{E} \left\{ \sum_{t=1}^{n} \ell^v(X_t) \delta_{X_t}(A) \prod_{i=1}^{m} \ell^{b_i}(X_i) \right\}.$$

Therefore,

$$(I - B) \left[ \left\{ \int_A \ell^v(x) \pi(dx) \right\} \prod_i \left\{ \int \ell^{a_i}(x)\, \pi(dx) \right\}^{s_i} \right]$$

$$= \frac{n^{s'+1} - n(n-1) \cdots (n - s')}{n^{s'+1}} c_{\mathbf{s},\mathbf{m}^*} \left\{ \int_A \ell^v(x) \pi(dx) \right\} \prod_{i=1}^{p} [\mathbb{E}\{\ell^{a_i}(X)\}]^{s_i}$$

$$- \sum_{b_k : \sum_{k=1}^{s'} b_k = s - v} \mathcal{O}(n^{-1}) c_{\mathbf{s},\mathbf{m}} \sum_{t=1}^{m} \left[ \prod_{i \neq t} \mathbb{E}\{\ell^{b_i}(X)\} \right] \int_A \ell^{b_t + v}(x) \pi(dx)$$

$$- \sum_{m=1}^{s'-1} \ \sum_{b_k : \sum_{k=1}^{m} b_k = s - v} \mathcal{O}(n^{m - s' - 1}) c_{\mathbf{s},\mathbf{m}} \mathbb{E} \left\{ \sum_{t=1}^{n} \ell^v(X_t) \delta_{X_t}(A) \prod_{i=1}^{m} \ell^{b_i}(X_i) \right\}$$

$$= \frac{n^{s'+1} - n(n-1) \cdots (n - s')}{n^{s'+1}} c_{\mathbf{s},\mathbf{m}^*} \left\{ \int_A \ell^v(x) \pi(dx) \right\} \prod_{i=1}^{p} [\mathbb{E}\{\ell^{a_i}(X)\}]^{s_i}$$

$$- \sum_{b_k : \sum_{k=1}^{s'} b_k = s - v} \mathcal{O}(n^{-1}) c_{\mathbf{s},\mathbf{m}} \sum_{t=1}^{m} \left[ \prod_{i \neq t} \mathbb{E}\{\ell^{b_i}(X)\} \right] \int_A \ell^{b_t + v}(x) \pi(dx)$$

$$- \sum_{m=1}^{s'-1} \ \sum_{b_k : \sum_{k=1}^{m} b_k = s - v} \mathcal{O}(n^{m - s' - 1}) c_{\mathbf{s},\mathbf{m}} \sum_{t=1}^{m} \left[ \prod_{i \neq t} \mathbb{E}\{\ell^{b_i}(X)\} \right] \int_A \ell^{b_t + v}(x) \pi(dx)$$

$$- \sum_{m=1}^{s'-1} \ \sum_{b_k : \sum_{k=1}^{m} b_k = s - v} \mathcal{O}(n^{m - s'}) c_{\mathbf{s},\mathbf{m}} \left\{ \int_A \ell^v(x) \pi(dx) \right\} \prod_{i=1}^{m} \mathbb{E}\{\ell^{b_i}(X_i)\}$$

$$\in \mathcal{A}_s^1.$$

The last inlcusion follows from that fact that $\{n^{s'+1} - n(n-1) \cdots (n - s')\}/n^{s'+1} = \mathcal{O}(n^{-1})$ and the number of solutions to $\sum_{k=1}^{m} b_k = s - v$ does not depend on $n$ but depends on $s$.

Now we suppose (12) holds for $k$. Then we can set

$$(I - B)^k h(\pi) = \sum_{(\mathbf{a},\mathbf{s},v) \in \mathfrak{J}_s} \alpha'_{\mathbf{a},\mathbf{s},v} \left\{ \int_A \ell^v(x) \pi(dx) \right\} \prod_i \left\{ \int \ell^{a_i}(x)\, \pi(dx) \right\}^{s_i},$$

where $|\alpha'_{\mathbf{a},\mathbf{s},v}| \le C_k(s)n^{-k}$. Then for $k+1$, we have

$$(I-B)^{k+1}h(\pi) = \sum_{(\mathbf{a},\mathbf{s},v)\in\mathfrak{J}_s} \alpha'_{\mathbf{a},\mathbf{s},v}(I-B)\left\{\int_A \ell^v(x)\pi(dx)\right\}\prod_i\left\{\int \ell^{a_i}(x)\,\pi(dx)\right\}^{s_i}.$$

Since for all $\mathbf{a},\mathbf{s},v$ such that $(\mathbf{a},\mathbf{s},v)\in\mathfrak{J}_s$, $\left\{\int_A \ell^v(x)\pi(dx)\right\}\prod_i\left\{\int \ell^{a_i}(x)\,\pi(dx)\right\}^{s_i}\in\mathcal{A}_s^0$, we have $(I-B)\left\{\int_A \ell^v(x)\pi(dx)\right\}\prod_i\left\{\int \ell^{a_i}(x)\,\pi(dx)\right\}^{s_i}\in\mathcal{A}_s^1$, namely,

$$(I-B)\left\{\int_A \ell^v(x)\pi(dx)\right\}\prod_i\left\{\int \ell^{a_i}(x)\,\pi(dx)\right\}^{s_i}$$

$$= \sum_{(\mathbf{b},\mathbf{t},u)\in\mathfrak{J}_s} \alpha_{\mathbf{b},\mathbf{t},u}(\mathbf{a},\mathbf{s},v)\left\{\int_A \ell^u(x)\pi(dx)\right\}\prod_i\left\{\int \ell^{b_i}(x)\,\pi(dx)\right\}^{t_i},$$

where $|\alpha_{\mathbf{b},\mathbf{t},u}(\mathbf{a},\mathbf{s},v)| \le C_0(s)n^{-1}$. Therefore,

$$(I-B)^{k+1}h(\pi)$$

$$= \sum_{(\mathbf{a},\mathbf{s},v)\in\mathfrak{J}_s} \alpha'_{\mathbf{a},\mathbf{s},v}\sum_{(\mathbf{b},\mathbf{t},u)\in\mathfrak{J}_s} \alpha_{\mathbf{b},\mathbf{t},u}(\mathbf{a},\mathbf{s},v)\left\{\int_A \ell^u(x)\pi(dx)\right\}\prod_i\left\{\int \ell^{b_i}(x)\,\pi(dx)\right\}^{t_i}$$

$$\in\mathcal{A}_s^{k+1}.$$

$\square$

# B  Proof of Theorem 2

In order to prove Theorem 2, we first make some preliminary observations.

Let function $f$ defined on the simplex $\Delta_m = \{\mathbf{q}\in\mathbb{R}^m : q_j \ge 0, \sum_{j=1}^m q_j = 1\}$. Define the generalized Bernstein basis polynomials of degree $n$ as

$$b_{n,\boldsymbol{\nu}}(\mathbf{q}) = \binom{n}{\boldsymbol{\nu}}\mathbf{q}^{\boldsymbol{\nu}}.$$

**Lemma 3.** $\left|\sum_{\boldsymbol{\nu}\in\bar{\Delta}_m}(\boldsymbol{\nu}/n-\mathbf{q})^{\boldsymbol{\alpha}}b_{n,\boldsymbol{\nu}}(\mathbf{q})\right| \lesssim n^{-\|\boldsymbol{\alpha}\|_1/2}.$

*Proof of Lemma 3.* It suffices to show that $\left|\sum_{\boldsymbol{\nu}\in\bar{\Delta}_m}(\boldsymbol{\nu}-n\mathbf{q})^{\boldsymbol{\alpha}}b_{n,\boldsymbol{\nu}}(\mathbf{q})\right| \lesssim n^{\|\boldsymbol{\alpha}\|_1/2}$. Since $\mathbf{q}\in\Delta_m$, we treat $T_{n,\boldsymbol{\alpha}} \equiv \sum_{\boldsymbol{\nu}\in\bar{\Delta}_m}(\boldsymbol{\nu}-n\mathbf{q})^{\boldsymbol{\alpha}}b_{n,\boldsymbol{\nu}}(\mathbf{q})$ as a function of the variables $q_1,\cdots,q_{m-1}$. For any $\boldsymbol{\beta}\in\mathbb{N}^{m-1}$ such that $\|\boldsymbol{\beta}\|_1 = 1$, we let $\boldsymbol{\gamma} = \boldsymbol{\gamma}(\boldsymbol{\beta}) \equiv (\boldsymbol{\beta}^\top, 0)^\top$. Additionally, let $\boldsymbol{\theta} = (0,\cdots,0,1)^\top \in\mathbb{N}^m$. Since

$$\partial^{\boldsymbol{\beta}}(\boldsymbol{\nu}-n\mathbf{q})^{\boldsymbol{\alpha}} = -n\boldsymbol{\alpha}^{\boldsymbol{\gamma}}(\boldsymbol{\nu}-n\mathbf{q})^{\boldsymbol{\alpha}-\boldsymbol{\gamma}} + n\boldsymbol{\alpha}^{\boldsymbol{\theta}}(\boldsymbol{\nu}-n\mathbf{q})^{\boldsymbol{\alpha}-\boldsymbol{\theta}},$$

and

$$\partial^{\boldsymbol{\beta}}b_{n,\boldsymbol{\nu}}(\mathbf{q}) = \binom{n}{\boldsymbol{\nu}}(\boldsymbol{\nu}^{\boldsymbol{\gamma}}\mathbf{q}^{\boldsymbol{\nu}-\boldsymbol{\gamma}} - \boldsymbol{\nu}^{\boldsymbol{\theta}}\mathbf{q}^{\boldsymbol{\nu}-\boldsymbol{\theta}})$$

$$= b_{n,\boldsymbol{\nu}}(\mathbf{q})\left\{\frac{1}{\mathbf{q}^{\boldsymbol{\gamma}}}(\boldsymbol{\nu}-n\mathbf{q})^{\boldsymbol{\gamma}} - \frac{1}{\mathbf{q}^{\boldsymbol{\theta}}}(\boldsymbol{\nu}-n\mathbf{q})^{\boldsymbol{\theta}}\right\},$$

we have

$$\partial^{\boldsymbol{\beta}}T_{n,\boldsymbol{\alpha}} = \sum_{\boldsymbol{\nu}\in\bar{\Delta}_m}\partial^{\boldsymbol{\beta}}(\boldsymbol{\nu}-n\mathbf{q})^{\boldsymbol{\alpha}}b_{n,\boldsymbol{\nu}}(\mathbf{q}) + \sum_{\boldsymbol{\nu}\in\bar{\Delta}_m}(\boldsymbol{\nu}-n\mathbf{q})^{\boldsymbol{\alpha}}\partial^{\boldsymbol{\beta}}b_{n,\boldsymbol{\nu}}(\mathbf{q})$$

$$= -n\boldsymbol{\alpha}^{\boldsymbol{\gamma}}T_{n,\boldsymbol{\alpha}-\boldsymbol{\gamma}} + n\boldsymbol{\alpha}^{\boldsymbol{\theta}}T_{n,\boldsymbol{\alpha}-\boldsymbol{\theta}} + \frac{1}{\mathbf{q}^{\boldsymbol{\gamma}}}T_{n,\boldsymbol{\alpha}+\boldsymbol{\gamma}} - \frac{1}{\mathbf{q}^{\boldsymbol{\theta}}}T_{n,\boldsymbol{\alpha}+\boldsymbol{\theta}},$$

i.e.,

$$\mathbf{q}^{\boldsymbol{\gamma}}\partial^{\boldsymbol{\beta}}T_{n,\boldsymbol{\alpha}} = -n\boldsymbol{\alpha}^{\boldsymbol{\gamma}}\mathbf{q}^{\boldsymbol{\gamma}}T_{n,\boldsymbol{\alpha}-\boldsymbol{\gamma}} + n\boldsymbol{\alpha}^{\boldsymbol{\theta}}\mathbf{q}^{\boldsymbol{\gamma}}T_{n,\boldsymbol{\alpha}-\boldsymbol{\theta}} + T_{n,\boldsymbol{\alpha}+\boldsymbol{\gamma}} - \frac{\mathbf{q}^{\boldsymbol{\gamma}}}{\mathbf{q}^{\boldsymbol{\theta}}}T_{n,\boldsymbol{\alpha}+\boldsymbol{\theta}}.$$

By summing the above equation over $\boldsymbol{\beta} \in \mathbb{N}^{m-1}$ such that $\|\boldsymbol{\beta}\|_1 = 1$, we have

$$\sum_{\|\boldsymbol{\beta}\|_1=1} \mathbf{q}^{\boldsymbol{\gamma}} \partial^{\boldsymbol{\beta}} T_{n,\boldsymbol{\alpha}}$$

$$= -n \sum_{\|\boldsymbol{\beta}\|_1=1} \boldsymbol{\alpha}^{\boldsymbol{\gamma}} \mathbf{q}^{\boldsymbol{\gamma}} T_{n,\boldsymbol{\alpha}-\boldsymbol{\gamma}} + n\boldsymbol{\alpha}^{\boldsymbol{\theta}} \sum_{\|\boldsymbol{\beta}\|_1=1} \mathbf{q}^{\boldsymbol{\gamma}} T_{n,\boldsymbol{\alpha}-\boldsymbol{\theta}} + \sum_{\|\boldsymbol{\beta}\|_1=1} T_{n,\boldsymbol{\alpha}+\boldsymbol{\gamma}} - \frac{1-\mathbf{q}^{\boldsymbol{\theta}}}{\mathbf{q}^{\boldsymbol{\theta}}} T_{n,\boldsymbol{\alpha}+\boldsymbol{\theta}}$$

$$= -n \sum_{\|\boldsymbol{\beta}\|_1=1} \boldsymbol{\alpha}^{\boldsymbol{\gamma}} \mathbf{q}^{\boldsymbol{\gamma}} T_{n,\boldsymbol{\alpha}-\boldsymbol{\gamma}} + n\boldsymbol{\alpha}^{\boldsymbol{\theta}}(1-\mathbf{q}^{\boldsymbol{\theta}}) T_{n,\boldsymbol{\alpha}-\boldsymbol{\theta}} + \sum_{\|\boldsymbol{\beta}\|_1=1} T_{n,\boldsymbol{\alpha}+\boldsymbol{\gamma}} - \frac{1-\mathbf{q}^{\boldsymbol{\theta}}}{\mathbf{q}^{\boldsymbol{\theta}}} T_{n,\boldsymbol{\alpha}+\boldsymbol{\theta}}$$

$$= -n \sum_{\|\boldsymbol{\beta}\|_1=1} \boldsymbol{\alpha}^{\boldsymbol{\gamma}} \mathbf{q}^{\boldsymbol{\gamma}} T_{n,\boldsymbol{\alpha}-\boldsymbol{\gamma}} + n\boldsymbol{\alpha}^{\boldsymbol{\theta}}(1-\mathbf{q}^{\boldsymbol{\theta}}) T_{n,\boldsymbol{\alpha}-\boldsymbol{\theta}} - \frac{1}{\mathbf{q}^{\boldsymbol{\theta}}} T_{n,\boldsymbol{\alpha}+\boldsymbol{\theta}},$$

where the last equality follows from the fact that $\sum_{\|\boldsymbol{\beta}\|_1=1} T_{n,\boldsymbol{\alpha}+\boldsymbol{\gamma}} + T_{n,\boldsymbol{\alpha}+\boldsymbol{\theta}} = 0$.

Therefore, we have the following recurrence formula:

$$T_{n,\boldsymbol{\alpha}+\boldsymbol{\theta}} = -n\mathbf{q}^{\boldsymbol{\theta}} \sum_{\|\boldsymbol{\beta}\|_1=1} \boldsymbol{\alpha}^{\boldsymbol{\gamma}} \mathbf{q}^{\boldsymbol{\gamma}} T_{n,\boldsymbol{\alpha}-\boldsymbol{\gamma}} + n\boldsymbol{\alpha}^{\boldsymbol{\theta}} \mathbf{q}^{\boldsymbol{\theta}}(1-\mathbf{q}^{\boldsymbol{\theta}}) T_{n,\boldsymbol{\alpha}-\boldsymbol{\theta}} - \mathbf{q}^{\boldsymbol{\theta}} \sum_{\|\boldsymbol{\beta}\|_1=1} \mathbf{q}^{\boldsymbol{\gamma}} \partial^{\boldsymbol{\beta}} T_{n,\boldsymbol{\alpha}}, \quad (15)$$

$$T_{n,\boldsymbol{\alpha}+\boldsymbol{\gamma}} = \frac{\mathbf{q}^{\boldsymbol{\gamma}}}{\mathbf{q}^{\boldsymbol{\theta}}} T_{n,\boldsymbol{\alpha}+\boldsymbol{\theta}} + n\boldsymbol{\alpha}^{\boldsymbol{\gamma}} \mathbf{q}^{\boldsymbol{\gamma}} T_{n,\boldsymbol{\alpha}-\boldsymbol{\gamma}} - n\boldsymbol{\alpha}^{\boldsymbol{\theta}} \mathbf{q}^{\boldsymbol{\gamma}} T_{n,\boldsymbol{\alpha}-\boldsymbol{\theta}} + \mathbf{q}^{\boldsymbol{\gamma}} \partial^{\boldsymbol{\beta}} T_{n,\boldsymbol{\alpha}}. \quad (16)$$

Using the recurrence fomrula (15), (16) and the fact that $T_{n,(1,0,\cdots,0)^{\top}} = 0, T_{n,(2,0,\cdots,0)^{\top}} = nq_1(1-q_1), T_{n,(1,1,0,\cdots,0)^{\top}} = -nq_1 q_2, T_{n,\boldsymbol{\alpha}}$ has the following form by using induction:

$$T_{n,\boldsymbol{\alpha}} = \sum_{j=1}^{\lfloor \|\boldsymbol{\alpha}\|_1/2 \rfloor} n^j \left( \sum_{\boldsymbol{\eta} \leq \boldsymbol{\alpha}} c_{j,\boldsymbol{\eta}} \mathbf{q}^{\boldsymbol{\eta}} \right), \quad (17)$$

where $c_{j,\boldsymbol{\eta}}$ is independent of $n$. Then we can conclude that $|T_{n,\boldsymbol{\alpha}}| \lesssim n^{\lfloor \|\boldsymbol{\alpha}\|_1/2 \rfloor} \lesssim n^{\|\boldsymbol{\alpha}\|_1/2}$.

$\square$

*Proof of Lemma 1.* We prove the theorem by induction on $k$.

For $k = 1$, by Taylor's expansion, there exists $\boldsymbol{\xi} \in \Delta_m$ such that

$$f\left(\frac{\boldsymbol{\nu}}{n}\right) = f(\mathbf{q}) + \sum_{\|\boldsymbol{\alpha}\|_1=1} \partial^{\boldsymbol{\alpha}} f(\boldsymbol{\xi}) \left(\frac{\boldsymbol{\nu}}{n} - \mathbf{q}\right)^{\boldsymbol{\alpha}}.$$

Then we have

$$|B_n(f)(\mathbf{q}) - f(\mathbf{q})| = \left| \sum_{\boldsymbol{\nu} \in \bar{\Delta}_m} \left\{ f\left(\frac{\boldsymbol{\nu}}{n}\right) - f(\mathbf{q}) \right\} b_{n,\boldsymbol{\nu}}(\mathbf{q}) \right|$$

$$= \left| \sum_{\boldsymbol{\nu} \in \bar{\Delta}_m} \left\{ \sum_{\|\boldsymbol{\alpha}\|_1=1} \partial^{\boldsymbol{\alpha}} f(\boldsymbol{\xi}) \left(\frac{\boldsymbol{\nu}}{n} - \mathbf{q}\right)^{\boldsymbol{\alpha}} \right\} b_{n,\boldsymbol{\nu}}(\mathbf{q}) \right|$$

$$\leq \|f\|_{C^1(\Delta_m)} \sum_{\|\boldsymbol{\alpha}\|_1=1} \sum_{\boldsymbol{\nu} \in \bar{\Delta}_m} \left| \left(\frac{\boldsymbol{\nu}}{n} - \mathbf{q}\right)^{\boldsymbol{\alpha}} \right| b_{n,\boldsymbol{\nu}}(\mathbf{q})$$

$$\leq \|f\|_{C^1(\Delta_m)} \sum_{\|\boldsymbol{\alpha}\|_1=1} \left\{ \sum_{\boldsymbol{\nu} \in \bar{\Delta}_m} \left(\frac{\boldsymbol{\nu}}{n} - \mathbf{q}\right)^{2\boldsymbol{\alpha}} b_{n,\boldsymbol{\nu}}(\mathbf{q}) \right\}^{1/2}$$

$$\lesssim \|f\|_{C^1(\Delta_m)} \sum_{\|\boldsymbol{\alpha}\|_1=1} n^{-1/2}$$

$$\lesssim_m \|f\|_{C^1(\Delta_m)} n^{-1/2},$$

where the second inequality follows from Cauchy–Schwarz inequality, and the third inequality follows from Lemma 3.

Suppose the theorem holds up through $k$. Now we prove the theorem for $k+1$. For $k+1$, by Taylor's expansion, there exists $\boldsymbol{\xi} \in \Delta_m$ such that

$$f(\frac{\boldsymbol{\nu}}{n}) = f(\mathbf{q}) + \sum_{\|\boldsymbol{\alpha}\|_1=1}^{k} \frac{\partial^{\boldsymbol{\alpha}} f(\mathbf{q})}{\boldsymbol{\alpha}!}(\frac{\boldsymbol{\nu}}{n} - \mathbf{q})^{\boldsymbol{\alpha}} + \sum_{\|\boldsymbol{\alpha}\|_1=k+1} \frac{\partial^{\boldsymbol{\alpha}} f(\boldsymbol{\xi})}{\boldsymbol{\alpha}!}(\frac{\boldsymbol{\nu}}{n} - \mathbf{q})^{\boldsymbol{\alpha}}.$$

Then we have

$$B_n(f)(\mathbf{q}) - f(\mathbf{q}) = \sum_{\boldsymbol{\nu} \in \bar{\Delta}_m} \left( f(\frac{\boldsymbol{\nu}}{n}) - f(\mathbf{q}) \right) b_{n,\boldsymbol{\nu}}(\mathbf{q})$$

$$= \sum_{\boldsymbol{\nu} \in \bar{\Delta}_m} \left\{ \sum_{\|\boldsymbol{\alpha}\|_1=1}^{k} \frac{\partial^{\boldsymbol{\alpha}} f(\mathbf{q})}{\boldsymbol{\alpha}!}(\frac{\boldsymbol{\nu}}{n} - \mathbf{q})^{\boldsymbol{\alpha}} + \sum_{\|\boldsymbol{\alpha}\|_1=k+1} \frac{\partial^{\boldsymbol{\alpha}} f(\boldsymbol{\xi})}{\boldsymbol{\alpha}!}(\frac{\boldsymbol{\nu}}{n} - \mathbf{q})^{\boldsymbol{\alpha}} \right\} b_{n,\boldsymbol{\nu}}(\mathbf{q})$$

$$= \sum_{\|\boldsymbol{\alpha}\|_1=1}^{k} \frac{\partial^{\boldsymbol{\alpha}} f(\mathbf{q})}{\boldsymbol{\alpha}!} \left\{ \sum_{\boldsymbol{\nu} \in \bar{\Delta}_m} (\frac{\boldsymbol{\nu}}{n} - \mathbf{q})^{\boldsymbol{\alpha}} b_{n,\boldsymbol{\nu}}(\mathbf{q}) \right\}$$

$$+ \sum_{\|\boldsymbol{\alpha}\|_1=k+1} \frac{\partial^{\boldsymbol{\alpha}} f(\boldsymbol{\xi})}{\boldsymbol{\alpha}!} \left\{ \sum_{\boldsymbol{\nu} \in \bar{\Delta}_m} (\frac{\boldsymbol{\nu}}{n} - \mathbf{q})^{\boldsymbol{\alpha}} b_{n,\boldsymbol{\nu}}(\mathbf{q}) \right\}.$$

Therefore,

$$(B_n - I)^{\lceil (k+1)/2 \rceil}(f)(\mathbf{q}) = \sum_{\|\boldsymbol{\alpha}\|_1=1}^{k} (B_n - I)^{\lceil (k+1)/2 \rceil - 1} \left\{ \frac{\partial^{\boldsymbol{\alpha}} f(\mathbf{q})}{\boldsymbol{\alpha}!} \sum_{\boldsymbol{\nu} \in \bar{\Delta}_m} (\frac{\boldsymbol{\nu}}{n} - \mathbf{q})^{\boldsymbol{\alpha}} b_{n,\boldsymbol{\nu}}(\mathbf{q}) \right\}$$

$$+ \sum_{\|\boldsymbol{\alpha}\|_1=k+1} (B_n - I)^{\lceil (k+1)/2 \rceil - 1} \left\{ \frac{\partial^{\boldsymbol{\alpha}} f(\boldsymbol{\xi})}{\boldsymbol{\alpha}!} \sum_{\boldsymbol{\nu} \in \bar{\Delta}_m} (\frac{\boldsymbol{\nu}}{n} - \mathbf{q})^{\boldsymbol{\alpha}} b_{n,\boldsymbol{\nu}}(\mathbf{q}) \right\}.$$
(18)

First, we consider the first term of the right-hand side of (18). We know that $(\boldsymbol{\alpha}!)^{-1} \partial^{\boldsymbol{\alpha}} f(\mathbf{q}) \sum_{\boldsymbol{\nu} \in \bar{\Delta}_m} (\boldsymbol{\nu}/n - \mathbf{q})^{\boldsymbol{\alpha}} b_{n,\boldsymbol{\nu}}(\mathbf{q}) \in C^{k+1-\|\boldsymbol{\alpha}\|_1}(\Delta_m)|$ since $f \in C^{k+1}(\Delta_m)$. By the induction hypothesis, we have

$$\left\| (B_n - I)^{\lceil (k+1-\|\boldsymbol{\alpha}\|_1)/2 \rceil} \left\{ \frac{\partial^{\boldsymbol{\alpha}} f(\mathbf{q})}{\boldsymbol{\alpha}!} \sum_{\boldsymbol{\nu} \in \bar{\Delta}_m} (\frac{\boldsymbol{\nu}}{n} - \mathbf{q})^{\boldsymbol{\alpha}} b_{n,\boldsymbol{\nu}}(\mathbf{q}) \right\} \right\|_{\infty}$$

$$\lesssim_{k+1-\|\boldsymbol{\alpha}\|_1,m} \left\| \frac{\partial^{\boldsymbol{\alpha}} f(\mathbf{q})}{\boldsymbol{\alpha}!} \sum_{\boldsymbol{\nu} \in \bar{\Delta}_m} (\frac{\boldsymbol{\nu}}{n} - \mathbf{q})^{\boldsymbol{\alpha}} b_{n,\boldsymbol{\nu}}(\mathbf{q}) \right\|_{C^{k+1-\|\boldsymbol{\alpha}\|_1}(\Delta_m)} n^{-(k+1-\|\boldsymbol{\alpha}\|_1)/2}.$$

Let

$$g_{\boldsymbol{\alpha}}(\mathbf{q}) = \frac{\partial^{\boldsymbol{\alpha}} f(\mathbf{q})}{\boldsymbol{\alpha}!} \sum_{\boldsymbol{\nu} \in \bar{\Delta}_m} (\frac{\boldsymbol{\nu}}{n} - \mathbf{q})^{\boldsymbol{\alpha}} b_{n,\boldsymbol{\nu}}(\mathbf{q}),$$

For any $|\boldsymbol{\beta}| \leq k+1 - \|\boldsymbol{\alpha}\|_1$, we have

$$\|\partial^{\boldsymbol{\beta}} g_{\boldsymbol{\alpha}}(\mathbf{q})\|_{\infty} = \left\| \frac{1}{\boldsymbol{\alpha}!} \sum_{0 \leq \boldsymbol{\gamma} \leq \boldsymbol{\beta}} \binom{\boldsymbol{\beta}}{\boldsymbol{\gamma}} \partial^{\boldsymbol{\alpha}+\boldsymbol{\gamma}} f(\mathbf{q}) \partial^{\boldsymbol{\beta}-\boldsymbol{\gamma}} \left\{ \sum_{\boldsymbol{\nu} \in \bar{\Delta}_m} (\frac{\boldsymbol{\nu}}{n} - \mathbf{q})^{\boldsymbol{\alpha}} b_{n,\boldsymbol{\nu}}(\mathbf{q}) \right\} \right\|_{\infty}$$

$$\lesssim_{k+1} \|f\|_{C^{k+1}(\Delta_m)} \sum_{0 \leq \boldsymbol{\gamma} \leq \boldsymbol{\beta}} \binom{\boldsymbol{\beta}}{\boldsymbol{\gamma}} \left\| \partial^{\boldsymbol{\beta}-\boldsymbol{\gamma}} \left\{ \sum_{\boldsymbol{\nu} \in \bar{\Delta}_m} (\frac{\boldsymbol{\nu}}{n} - \mathbf{q})^{\boldsymbol{\alpha}} b_{n,\boldsymbol{\nu}}(\mathbf{q}) \right\} \right\|_{\infty}$$

$$\lesssim_{k+1} \|f\|_{C^{k+1}(\Delta_m)} n^{-\|\boldsymbol{\alpha}\|_1/2},$$

where the last inequality follows from the fact that $\left\|\partial^{\boldsymbol{\beta}-\boldsymbol{\gamma}}\left\{\sum_{\boldsymbol{\nu}\in\bar{\Delta}_m}(\boldsymbol{\nu}/n-\mathbf{q})^{\boldsymbol{\alpha}}b_{n,\boldsymbol{\nu}}(\mathbf{q})\right\}\right\|_\infty \lesssim$ $n^{-\|\boldsymbol{\alpha}\|_1/2}$ which can be derived by using the form of $T_{n,\boldsymbol{\alpha}}$ in (17).

Therefore, we have

$$\left\|(B_n-I)^{\lceil(k+1-\|\boldsymbol{\alpha}\|_1)/2\rceil}\left\{\frac{\partial^{\boldsymbol{\alpha}}f(\mathbf{q})}{\boldsymbol{\alpha}!}\sum_{\boldsymbol{\nu}\in\bar{\Delta}_m}(\frac{\boldsymbol{\nu}}{n}-\mathbf{q})^{\boldsymbol{\alpha}}b_{n,\boldsymbol{\nu}}(\mathbf{q})\right\}\right\|_\infty$$
$$\lesssim_{k+1}\|f\|_{C^{k+1}(\Delta_m)}n^{-(k+1)/2}.$$

Then we consider the second term of the right-hand side of (18).

$$\left\|\sum_{\|\boldsymbol{\alpha}\|_1=k+1}(B_n-I)^{\lceil(k+1)/2\rceil-1}\left\{\frac{\partial^{\boldsymbol{\alpha}}f(\boldsymbol{\xi})}{\boldsymbol{\alpha}!}\sum_{\boldsymbol{\nu}\in\bar{\Delta}_m}(\frac{\boldsymbol{\nu}}{n}-\mathbf{q})^{\boldsymbol{\alpha}}b_{n,\boldsymbol{\nu}}(\mathbf{q})\right\}\right\|_\infty$$
$$\lesssim_{k+1}\|(B_n-I)^{\lceil(k+1)/2\rceil-1}\|_\infty\|f\|_{C^{k+1}(\Delta_m)}\sum_{\|\boldsymbol{\alpha}\|_1=k+1}\sum_{\boldsymbol{\nu}\in\bar{\Delta}_m}\left|(\frac{\boldsymbol{\nu}}{n}-\mathbf{q})^{\boldsymbol{\alpha}}\right|b_{n,\boldsymbol{\nu}}(\mathbf{q})$$
$$\lesssim_{k+1}\|(B_n-I)^{\lceil(k+1)/2\rceil-1}\|_\infty\|f\|_{C^{k+1}(\Delta_m)}\sum_{\|\boldsymbol{\alpha}\|_1=k+1}\left\{\sum_{\boldsymbol{\nu}\in\bar{\Delta}_m}(\frac{\boldsymbol{\nu}}{n}-\mathbf{q})^{2\boldsymbol{\alpha}}b_{n,\boldsymbol{\nu}}(\mathbf{q})\right\}^{1/2}$$
$$\lesssim_{k+1,m}\|(B_n-I)^{\lceil(k+1)/2\rceil-1}\|_\infty\|f\|_{C^{k+1}(\Delta_m)}n^{-(k+1)/2}.$$

Finally, we have

$$\|(B_n-I)^{\lceil(k+1)/2\rceil}(f)(\mathbf{q})\|_\infty \le \sum_{\|\boldsymbol{\alpha}\|_1=1}^{k}\left\|(B_n-I)^{\lceil(k+1)/2\rceil-1}\left\{\frac{\partial^{\boldsymbol{\alpha}}f(\mathbf{q})}{\boldsymbol{\alpha}!}\sum_{\boldsymbol{\nu}\in\bar{\Delta}_m}(\frac{\boldsymbol{\nu}}{n}-\mathbf{q})^{\boldsymbol{\alpha}}b_{n,\boldsymbol{\nu}}(\mathbf{q})\right\}\right\|_\infty$$
$$+\left\|\sum_{\|\boldsymbol{\alpha}\|_1=k+1}(B_n-I)^{\lceil(k+1)/2\rceil-1}\left\{\frac{\partial^{\boldsymbol{\alpha}}f(\boldsymbol{\xi})}{\boldsymbol{\alpha}!}\sum_{\boldsymbol{\nu}\in\bar{\Delta}_m}(\frac{\boldsymbol{\nu}}{n}-\mathbf{q})^{\boldsymbol{\alpha}}b_{n,\boldsymbol{\nu}}(\mathbf{q})\right\}\right\|_\infty$$
$$\lesssim_{k+1,m}\sum_{\|\boldsymbol{\alpha}\|_1=1}^{k}\|(B_n-I)^{\lceil\|\boldsymbol{\alpha}\|_1/2\rceil-1}\|_\infty\|f\|_{C^{k+1}(\Delta_m)}n^{-(k+1)/2}$$
$$+\|(B_n-I)^{\lceil(k+1)/2\rceil-1}\|_\infty\|f\|_{C^{k+1}(\Delta_m)}n^{-(k+1)/2}$$
$$\lesssim_{k+1,m}\|f\|_{C^{k+1}(\Delta_m)}n^{-(k+1)/2}.$$

The last inequality holds when $\|B_n-I\|_\infty$ is bounded by a constant independent of $n$. In fact, $\|(B_n-I)f\|_\infty = \sup_{\mathbf{q}\in\Delta_m}|(B_n-I)f(\mathbf{q})| = \sup_{\mathbf{q}\in\Delta_m}\left|\sum_{\boldsymbol{\nu}\in\bar{\Delta}_m}\{f(\boldsymbol{\nu}/n)-f(\mathbf{q})\}b_{n,\boldsymbol{\nu}}(\mathbf{q})\right| \le 2\|f\|_\infty$ and $\|B_n-I\|_\infty = \sup_{f\in C^k(\Delta_m),\|f\|_\infty\le 1}\|(B_n-I)f\|_\infty \le \sup_{f\in C^k(\Delta_m),\|f\|_\infty\le 1}2\|f\|_\infty \le 2$. $\square$

*Proof of Theorem 2.* The first claim follows from Lemma 1 and the fact that $\max_{s\in[m]}\|g_s\|_{C^{2k}(\Delta_m)} \le G$ and $\mathbb{E}_{X^n}\{D_{n,k}(g_s)(\mathbf{T}/n)\} = C_{n,k}(g_s)(\mathbf{q})$.

Additionally, we have

$$D_{n,k}(g_s)(\mathbf{q}) = \sum_{j=0}^{k-1}\binom{k}{j+1}(-1)^j B_n^j(g_s)(\mathbf{q})$$
$$= \sum_{j=0}^{k-1}\binom{k}{j+1}(-1)^j\left\{g_s(\mathbf{q})+\mathcal{O}_{k,m,G}(n^{-1})\right\}$$
$$= g_s(\mathbf{q})+\mathcal{O}_{k,m,G}(n^{-1}),$$

and

$$\mathbb{E}_{X^n}\left[\{D_{n,k}(g_s)(\mathbf{T}/n)\}^2\right] = \sum_{\boldsymbol{\nu}\in\bar{\Delta}_m} \{D_{n,k}(g_s)(\boldsymbol{\nu}/n)\}^2 \, b_{n,\boldsymbol{\nu}}(\mathbf{q})$$

$$= B_n\left[\{D_{n,k}(g_s)\}^2\right](\mathbf{q})$$

$$= \{D_{n,k}(g_s)(\mathbf{q})\}^2 + \mathcal{O}_{k,m,G}(n^{-1})$$

$$= \left\{g_s(\mathbf{q}) + \mathcal{O}_{k,m,G}(n^{-1})\right\}^2 + \mathcal{O}_{k,m,G}(n^{-1})$$

$$= g_s^2(\mathbf{q}) + \mathcal{O}_{k,m,G}(n^{-1}).$$

Therefore,

$$\mathrm{Var}_{X^n}\{D_{n,k}(g_s)(\mathbf{T}/n)\} = \mathbb{E}_{X^n}\left[\{D_{n,k}(g_s)(\mathbf{T}/n)\}^2\right] - [\mathbb{E}_{X^n}\{D_{n,k}(g_s)(\mathbf{T}/n)\}]^2$$

$$= g_s^2(\mathbf{q}) + \mathcal{O}_{k,m,G}(n^{-1}) - \left\{g_s(\mathbf{q}) + \mathcal{O}_{k,m}(n^{-k})\right\}^2$$

$$= \mathcal{O}_{k,m,G}(n^{-1}).$$

$\square$

## C  Proof of Theorem 3

*Proof of Theorem 3.* By Theorem 2, it suffices to let $\widetilde{P}_{X^n}(x = u_s | y = y^*) = D_{n,k}(g_s)(\mathbf{T}/n)$. Moreover, we have $\sum_{s=1}^{m} D_{n,k}(g_s)(\mathbf{T}/n) = 1$. $\square$

