# OpenReview forum: "A Black-Box Debiasing Framework for Conditional Sampling"
_NeurIPS.cc/2025/Conference — NeurIPS 2025 poster_

### Official Review · Reviewer_FYyK · 2025-06-22

**Clarity:** 3
**Significance:** 3
**Originality:** 3
**Rating:** 4
**Confidence:** 3

**Summary:**

This paper studies the task of sampling from the posterior distribution $P _ {X|Y=y^*}$ given i.i.d. samples $X _ {1:n}$ from an unknown prior distribution $\pi$ and a known likelihood model $P _ {Y|X}$. The authors propose a novel approach to reduce the bias of the plug-in approximation (i.e., replacing $\pi$ with the empirical distribution of the samples) so that the TV distance of the averaged posterior distribution (over the randomness of $X _ {1:n}$) to the true posterior distribution can decay at $O(n^{-k})$ for any integer $k$, while the average variance of the posterior distribution remains $O(n^{-1})$. The authors prove two cases of the continuous and discrete prior distributions, respectively, and validate their theoretical findings through numerical experiments on Bernoulli prior distribution.

**Questions:**

See weakness part above.

**Ethical Concerns:**

["NO or VERY MINOR ethics concerns only"]

**Final Justification:**

The authors have addressed most of the concerns from me and the other reviewers during the rebuttal. While I believe the paper could still benefit from enhanced experimental results, the current form is enough for getting an acceptance, especially considering the theoretical contributions. I thus maintain my original evaluation.

**Limitations:**

Yes.

**Quality:**

3

**Strengths And Weaknesses:**

**Strengths**

This paper is theoretically solid and well-written. I have gone through the proofs and have not found any significant issues. The notations are mostly clear, though I had some difficulty understanding the definition of $B _ n$ and $B _ n^j$ at first read (but it was clarified in the proof and the pseudocodes). The mathematical derivations are rigorous, and the assumptions are clearly stated. The main results are also clearly presented, with an easy-to-follow sketch of proof in the main text providing good intuition.

**Weaknesses**

Writings can be improved in some places. For example, in typical machine learning conference papers, it is not very common to include heavy mathematical details in the abstract and introduction sections. For instance, content between lines 49 and 71 can be briefly summarized in the introduction but elaborated in the main body. The contribution part can also be summarized and itemized for better clarity. Also, the authors can convert some inline equations to display style for better readability (e.g., lines 385 to 389). I suggest the authors to refer to some well-written papers in top ML conferences for better writing styles.

While I understand that this is a statistical theory paper, as the authors propose theoretically-grounded improved methodologies, the empirical results can be further strengthened, such as adding experiments on continuous prior distributions and trying estimators with bias $O(n^{-k})$ for some larger $k$. The rejection sampling proposed in Alg. 2 can also be tried. As the authors mentioned, the proposed framework can be applied to generative models like diffusion models and VAEs for conditional sampling, it would significantly enhance the impact of this work if the authors can demonstrate the effectiveness of their methods on some real-world datasets (i.e., with more complex prior distributions).

---

> ### Author Rebuttal · Authors · 2025-07-30
>
> **Q1:**  Writings can be improved in some places.
>
> **Answer.**  We appreciate the reviewer’s helpful suggestions on the writing.
>     We will revise the abstract and introduction by summarizing key ideas and deferring mathematical details to the main body.
>     We will also rewrite the contributions into a clear, itemized form for better clarity.
>     In addition, we have converted several inline equations to display style.
>
>
> **Q2:** Add experiments.
>
> **Answer.** We have added additional experiments to evaluate our method in a more complex scenario and also conducted additioanl experiments for Gaussian/Bernoulli setting when $k=3$ and $4$.
>     Please refer to our detailed response to Reviewer yaah’s Question 1 and 2.
>
>
> **Q3:** As the authors mentioned, the proposed framework can be applied to generative models like diffusion models and VAEs for conditional sampling, it would significantly enhance the impact of this work if the authors can demonstrate the effectiveness of their methods on some real-world datasets (i.e., with more complex prior distributions).
>
> **Answer.** We have included a more complex experiment in our response to Reviewer yaah's Question 1 and such additive Gaussian noise model setting was previously considered in the conditional generative model literature, e.g. ref. [22] in our paper.
>     Please refer to our response to Reviewer LUxm's Question 4.

---

> > ### Comment · Reviewer_FYyK · 2025-08-01
> >
> > I thank the authors for carefully providing response to all questions and concerns raised in my review, and hope that the authors could reorganize the writing of the paper at camera-ready (also please make sure the abstract is displayed correctly on OpenReview). I have also checked your additional experimental result, which looks nice even without plots showing the trend.
> >
> > That said, I still believe that the paper has room for significant refinement, such as the experiments on more complex datasets and pretrained models, but the current form is enough for getting an acceptance. I will therefore maintain my original score.

---

> > > ### Author Response · Authors · 2025-08-08
> > >
> > > Thank you for your careful reading of our paper and for acknowledging our detailed responses to your concerns. We appreciate your feedback, which helps us improve the clarity of our work. If the paper is accepted, we will update it to address the issues you noted such as fixing the abstract display and incorporating the added experiments for $k=3,4$ as well as for the Gaussian mixture distribution prior case.
> > >
> > > Regarding the experiments on more complex datasets, we plan to further explore the ImageNet example, similar to I.2 in "Reward-Directed Conditional Diffusion: Provable Distribution Estimation and Reward Improvement". For a specific downstream task such as expected reward estimation, the relevant dimension is task-dependent and can be even smaller than the intrinsic dimension of ImageNet. In such cases, we anticipate that our debiasing method may help reduce bias when a reasonable amount of training data is available.

---

### Official Review · Reviewer_LUxm · 2025-07-01

**Clarity:** 3
**Significance:** 2
**Originality:** 3
**Rating:** 4
**Confidence:** 3

**Summary:**

Let $(X,Y)$ be a joint distribution, with the likelihood $P_{Y|X}$ assumed known. The paper tackles the task of estimating the posterior $P_{X\mid Y = y^*}$ from i.i.d. samples $X_1, \dots, X_n$ drawn from the joint. It concentrates on the **bias** of the estimator, measured as the total-variation distance between the *average* of the estimators and the true posterior. The key technical idea is to apply an operator—borrowed from polynomial-approximation/Richardson-extrapolation ideas—to build an estimator whose bias shrinks as $O(n^{-k})$ while the variance stays at the usual $O(n^{-1})$, for any fixed constant $k$.

**Questions:**

See weakness section.

**Ethical Concerns:**

["NO or VERY MINOR ethics concerns only"]

**Final Justification:**

Though I'm still skeptical about the practical significance of the work in machine learning, it is nevertheless a solid contribution. I therefore maintain my current rating, favoring the acceptance of the paper.

**Limitations:**

yes

**Quality:**

3

**Strengths And Weaknesses:**

**Strengths**

1. As far as I can tell, the operator-based approach is elegant and may be applicable in a broader context. Although similar ideas might exist (or even be well-known) in approximation theory, their use in the current setting appears novel.
2. The results are non-trivial, and the technical proofs are carefully worked out. I did not spot any significant mistakes.

**Weaknesses**

1. My main reservation of the paper is its objective: eliminating the **bias** of the estimator is rather weak. For example, for a single sample $X$ from a distribution $p$ the empirical distribution $\delta_X$ is completely unbiased. A more relevant metric is the **expected risk** that compares the actual estimate to the underlying truth before taking the outer expectation. Given the variance bound $O(n^{-1})$, the expected risk of the proposed estimator shows no improvement over the naïve plug-in estimator.  Reducing bias relative to the plug-in estimator is not as surprising as the authors suggest.
2. The guarantees are essentially asymptotic—the constant hidden in the $O(\cdot)$ notation depends on $k$. A quick check of the proofs suggests it can be as large as $2^k$, which further limits practical usefulness. Can the authors state the constant explicitly, or at least its order?
3. The assumption that the likelihood $\ell(x)$ is bounded away from zero feels unnatural and needs justification.
4. In my opinion, this is a classical statistics paper packaged as a machine-learning contribution. I can hardly imagine how the claimed application scenarios (e.g., generative models) match the actual results. A concrete, convincing example would help.

**Overall**

Despite these criticisms, the paper solves a non-trivial problem in a niche area, and the work is solid. I therefore recommend borderline accept.

---

> ### Author Rebuttal · Authors · 2025-07-30
>
> **Q1:**  My main reservation of the paper is its objective: eliminating the bias of the estimator is rather weak. For example, for a single sample $X$ from a distribution $p$ the empirical distribution $\delta_X$ is completely unbiased. A more relevant metric is the expected risk that compares the actual estimate to the underlying truth before taking the outer expectation. Given the variance bound $\mathcal{O}(n^{-1})$, the expected risk of the proposed estimator shows no improvement over the naive plug-in estimator. Reducing bias relative to the plug-in estimator is not as surprising as the authors suggest.
>
> **Answer.**  We appreciate the reviewer’s comment and agree that bias alone is not the only relevant criterion.
>
> However, in many problems in statistics and machine learning, it is useful to reduce the bias, even if the variance remains the same order.
>     For example, "Debiasing Evidence Approximations: On Importance-weighted Autoencoders and Jackknife Variational Inference" by Nowozin (2018, ICML) demonstrates the practical importance of bias reduction.
>     In his paper, the jackknife technique is used to reduce bias in estimating the log-marginal likelihood, even though it does not improve the variance rate.
>     Our contribution follows a similar motivation.
>     While the variance of the proposed estimator remains at order $\mathcal{O}(n^{-1})$, we show that its bias decays faster than that of the standard plug-in estimator.
>
>
> **Q2:** The guarantees are essentially asymptotic—the constant hidden in the  $\mathcal{O}(\cdot)$ notation depends on $k$. A quick check of the proofs suggests it can be as large as $2^k$, which further limits practical usefulness. Can the authors state the constant explicitly, or at least its order?
>
> **Answer.**  The constant hidden in the bound $\mathcal{O}\_{L\_1, L\_2, k}(n^{-k})$ can be written explicitly in the form $C\cdot \big((L\_2-L\_1)^2/2L\_1^2\big)^k k!$ where   $C>0$ does not depend on $k$.
>     Although this hidden constant does limit the practical value for large $k$, the result is still meaningful for moderate values such as $k=2,3,4,5$ and helps provide theoretical insight into the asymptotic behavior.
>
>
> **Q3:** The assumption that the likelihood  $\ell(x)$ is bounded away from zero feels unnatural and needs justification.
>
> **Answer.** We agree that the assumption that the likelihood $\ell(x)$ is bounded away from zero may appear strong.
>     However, in many structured settings, such a lower bound can be justified with some more mild conditions.
>
> For example, consider the Reward-Directed Conditional Diffusion model setup in Yuan et al. (2023, NeurIPS) that $y=f^\*(x)+\xi$ where $\xi\sim\mathcal{N}(0,\sigma^2)$ and the data sampling distribution $P\_x$ is supported on a low-dimensional linear subspace, i.e., $x=Az$. They further assume that $z\sim(0, \Sigma)$ and $\lambda_{\min} I\_d \preceq \Sigma \preceq \lambda\_{\max} I_d$.
>     If we further assume that the distribution of $x$ has compact support, we can ensure that the likelihood is bounded away from zero for any given $y^*$.
>
> **Q4:** In my opinion, this is a classical statistics paper packaged as a machine-learning contribution. I can hardly imagine how the claimed application scenarios (e.g., generative models) match the actual results. A concrete, convincing example would help.
>
> **Answer.** We have included a more sophisticated experiment in our response to Reviewer yaah's Question 1 and such additive Gaussian noise model setting was previously considered in the conditional generative model literature, e.g. ref. [22] in our paper.
>     Moreover, prior work "Debiasing Evidence Approximations: On Importance-weighted Autoencoders and Jackknife Variational Inference" by Nowozin (2018, ICML) has successfully applied jackknife technique to reduce the bias in estimating the log-marginal likelihood to real datasets like MNIST.
>     Their experiment suggests that for real data, there could be an intrinsic dimension much samller than the ambient dimension, so that the curse of dimensionlity may not be present.

---

> > ### Comment · Reviewer_LUxm · 2025-08-02
> >
> > I thank the authors for their responses to my questions, which clarified some of my concerns. Though I'm still skeptical about the practical significance of the work in machine learning, it is nevertheless a solid contribution. I therefore maintain my current rating, favoring the acceptance of the paper.

---

> > > ### Author Response · Authors · 2025-08-08
> > >
> > > Thank you for your encouraging feedback and for the time you dedicated to reviewing our work. If the paper is accepted, we will update it by incorporating the added experiments for $k=3,4$ as well as for the Gaussian mixture distribution prior case. Regarding the curse of dimensionality, We will also include the analysis discussed in the rebuttal of the dependence on $k$ and the dimension $d$.
> > >
> > > Regarding the applicability to real datasets, we plan to further explore the ImageNet example, similar to I.2 in "Reward-Directed Conditional Diffusion: Provable Distribution Estimation and Reward Improvement". For a specific downstream task such as expected reward estimation, the relevant dimension is task-dependent and can be even smaller than the intrinsic dimension of ImageNet. In such cases, we anticipate that our debiasing method may help reduce bias when a reasonable amount of training data is available.

---

### Official Review · Reviewer_yaah · 2025-07-03

**Clarity:** 3
**Significance:** 2
**Originality:** 3
**Rating:** 4
**Confidence:** 3

**Summary:**

This paper examines the problem of taking a sample of unconditional observations $X$ and a known data likelihood distribution $P_{Y | X} $ and estimating the posterior distribution $P(X | Y = y^\ast)$. The approach in this work does not estimate a parametric function for the distribution of $X$ but instead estimates the posterior distribution using a weighted sum of empirical dirac-delta distributions obtained by an iterative data resampling process. The first main result is to present the form of the weighted sum of iterative resampling procedures and to prove that this approximates the true posterior distribution with an approximation error of $\mathcal{O}(n ^{-k})$ where $n$ is the sample size and $k$ is the number of resampling iterations. Next, a similar theorem with simpler assumptions is proved in the case where $X$ takes on a finite number of discrete values. Two algorithms are presented: one for approximating the posterior distribution using a weighted sum of empirical distribution for $k$ rounds of iterative resampling, and one for drawing a single sample from the approximate posterior in the case where $X$ is discrete. An experimental section validates the theoretical claims in the case where $X$ is a Bernoulli distribution and $Y | X$ follows a Gaussian distribution where the mean depends on $X$ using $k = 2$ resamplings.

**Questions:**

* Can more experiments be added for more complex scenarios? How well does this method scale with dimension and distributional complexity?
* How does the Gaussian/Bernoulli experiment unfold for $k > 2$? Is there eventually a limit to the benefit of increasing $k$? The theoretical results appear to show that increasing $k$ should provide increasing benefits, but I am skeptical that iterative resampling can really provide increasingly accurate estimates since no "new" information is really being generated.
* Can the authors provide an intuitive explanation for the form of $D_{n, k}$? Even after reading through the proof of Theorem 1 a few times, it is difficult to understand why the form $D_{n, k}$ enables better approximation accuracy than the naive plug-in estimator.
* Does this method overcome limitations of the empirical Dirac distribution in high dimensional settings?

**Ethical Concerns:**

["NO or VERY MINOR ethics concerns only"]

**Final Justification:**

The authors provided additional experiments using more complex distributions and provided additional exploration into the effect of $k$. I am not convinced that this method is practical for high-dimensional machine learning situations. Nonetheless, the statistical contribution of this paper seems solid. I weakly recommend accepting the paper.

**Limitations:**

Limitations and societal consequences are not discussed by the authors. Lack of discussion about societal consequences does not impact my view of the work. Lack of discussion about limitations does impact my view, in particular limitations that relate to the scalability of the proposed method in terms of distribution dimension and complexity. The method is only applied in a very simple scenario, which makes me think that these limitations might be significant.

**Paper Formatting Concerns:**

There are numerous spelling errors throughout the paper which should be corrected.

**Quality:**

2

**Strengths And Weaknesses:**

**Strengths**
* The goal of estimating posterior distributions using only unconditional samples and a known likelihood is an important and relevant problem. Providing a method for obtaining better approximations that the naive (and biased) plug-in empirical estimate that only requires straightforward iterative resampling and weighting plug-in estimates has the potential to be a useful and lightweight solution.
* The level of technical theoretical details is high, and the paper proves several non-trivial results. The theoretical results appear valid from what I reviewed, but the details are very dense and time consuming to thoroughly validate.
* The experimental results in the Bernoulli/Gaussian setting match very well with the theoretical predictions for $k=2$ resamplings.

**Weaknesses**
* The experimental section is extremely limited and only examines a single very simple 1D situation. Furthermore, only $k=2$ resamplings are studied, while the theoretical results appear to demonstrate significant benefit for increasingly large $k$. It would be helpful to see experiments on more complex scenarios and for higher $k$.
* The method relies heavily on the empirical Dirac distribution. While this alleviates the difficulty of learning a parametric model, I would guess that it heavily limits the scalability of the proposed method. In particular, the method seems very susceptible to the curse of dimensionality and prohibitively large number of samples $n$ might be needed in high-dimensional cases. The paper mentions diffusion and other generative models a few times, but I am not sure whether the proposed method is really viable in those kinds of situations.

---

> ### Author Rebuttal · Authors · 2025-07-30
>
> **Q1:** Can more experiments be added for more complex scenarios? How well does this method scale with dimension and distributional complexity?
>
> **Answer.** Thank you for the thoughtful suggestion.
>     We have added an additional experiment to evaluate our method in a more complex scenario.
>     Specifically, we consider a setting where $X$ follows a Gaussian mixture distribution, that is,
>     $X\sim \frac{1}{2}\mathcal{N}(0,1)+\frac{1}{2}\mathcal{N}(1,1)$, and $Y=X+\xi$ where $\xi\sim\mathcal{N}(0, 1/16)$.
>     Additionally, we let $y^*=0.8$ and $A=\\{x: x\ge 0.5\\}$.
>     The additive Gaussian noise model was previously considered in the conditional generative model literature, e.g. ref. [22] in our paper.
>     We use this experiment to validate the theoretical rate $|\mathbb{E}\_{X^n}[D\_{n,k}f(\hat{\pi})(A)]-f(\pi)(A)|=\mathcal{O}(n^{-k})$.
>
> First, we compute the ground truth value $f(\pi)(A)\approx0.8795$.
>     Next, since $\mathbb{E}\_{X^n}[D\_{n,k}f(\hat{\pi})(A)]$ does not have a closed form, we approximate it using Monte Carlo simulation.
>     To ensure that the Monte Carlo error is negligible compared to the bias $\mathcal{O}(n^{-k})$, we choose the number of Monte Carlo samples $M$ such that $M\gg n^{2k-1}$.
>
> We have run the code for $k=1$ and $2$. Specifically, we choose $M=n^3$ for $k=1$ and $M=n^4$ for $k=2$.
>
> The experimental results for this setting are summarized in the following table, where each entry represents the logarithm of the absolute error between the Monte Carlo estimate and the true value.
>
> |       | n=10    | n=20    | n=30    | n=40    | n=50    | n=60    | n=70    | n=80    | n=90    | n=100   |
> |-------|---------|---------|---------|---------|---------|---------|---------|---------|---------|---------|
> | k=1   | -3.1113 | -4.1539 | -4.6486 | -4.9098 | -5.2122 | -5.3870 | -5.5418 | -5.6924 | -5.8139 | -5.9296 |
> | k=2   | -4.8037 | -5.8835 | -6.6880 | -7.1578 | -7.8573 | -8.0784 | -8.4541 | -8.7667 | -9.0330 | -9.2975 |
>
> When we plot these errors against $\log(n)$, the resulting lines for $k=1$ and $k=2$ are nearly parallel to the lines of $\mathcal{O}(n^{-1})$ and $\mathcal{O}(n^{-2})$ respectively.
>     This experiment demonstrates that the proposed method scales well in the presence of distributional complexity.
>
> In general, the curse of dimensionality may arise and depends on the specific distribution of $X$ and the likelihood function.
>     There is no universal relationship between $n$ and the dimension $d$.
>     However, to provide some intuition, we can give an example that illustrates how $n$ and $d$ may relate.
>
> Suppose that $Y=(Y(1),\\dots,Y(d))$ and $X=(X(1),\\dots, X(d))$ have i.i.d. components, and $L_1\le p(Y(i)|X(i)) \le L_2$ for $1\le i\le d$.
> Then we have $\ell(X):=p(Y|X)\in[L_1^d, L_2^d]$.
> Consider line 390 in our paper, the residual term can be rewritten explicitly as $C\cdot \big((L_2^d-L_1^d)^2/2L_1^{2d}\big)^kk!\mathcal{O}(n^{-k})$ where   $C>0$ does not depend on $k$ and $d$.
>     To ensure that this residual term is negligible compared to the final term retained in the Taylor expansion, it suffices to let $n$ and $d$ satisfy that $\big((L_2^d-L_1^d)^2/L_1^{2d}\big)^k\ll n^{1/2}$.
>     Therefore, by choosing $n$ such that $(L_2/L_1)^d\ll n^{1/4k}$, our method scales with dimension.
>
> **Q2:** How does the Gaussian/Bernoulli experiment unfold for $k>2$? Is there eventually a limit to the benefit of increasing $k$? The theoretical results appear to show that increasing $k$ should provide increasing benefits, but I am skeptical that iterative resampling can really provide increasingly accurate estimates since no "new" information is really being generated.
>
> **Answer.**  We also conducted additional experiments for Gaussian/Bernoulli setting when $k=3$ and $4$.
>     Under the binary prior setting, we have $\mathbb{E}\_{X^n}[B\_n^s(g)(T/n)]=B\_n^{s+1}(g)(q)$ for any $s\ge 0$, which enables us to compute the exact value of $\mathbb{E}\_{X^n}[D\_{n,k}(g)(T/n)]$ in closed form, that is, $\mathbb{E}\_{X^n}[D\_{n,k}(g)(T/n)]=\mathbb{E}\_{X^n}[D\_{n,k}(g)(T/n)]=\mathbb{E}\_{X^n}[\sum\_{j=0}^k\binom{k}{j+1}(-1)^jB\_n^j(g)(T/n)]=\sum_{j=0}^k\binom{k}{j+1}(-1)^jB\_n^{j+1}(g)(q)$.
>     This allows us to directly assess the approximation error without relying on Monte Carlo simulation.
>
> The experimental results for $k=3$ and $4$ are reported in the following table, where each entry represents the logarithm of the absolute error between the exact closed-form expression value and the true value.
>
> |       | n=100   | n=300   | n=500   | n=700   | n=900   |
> |-------|---------|---------|---------|---------|---------|
> | k=3   | -13.1934 | -16.6413 | -18.2059 | -19.2293 | -19.9910 |
> | k=4   | -15.3086 | -19.6794 | -21.7267 | -23.0750 | -24.0818 |
>
> When we plot these errors against $\log(n)$, the resulting lines for $k=3$ and $4$ are parallel to the lines of $\mathcal{O}(n^{-3})$ and $\mathcal{O}(n^{-4})$ respectively.
>
> There is no limit to the benefit of increasing $k$ for fixed $P(X,Y)$ (if the dimension $d$ increases with $n$, then we may require $kd \ll \log n$ in the example in our answer to Q1).
>     Although there is no "new" information, the iterative resampling can still provide increasingly accurate approximation.
>     For example, prior work such as "Debiasing Evidence Approximations: On Importance-weighted Autoencoders and Jackknife Variational Inference" by Nowozin (2018, ICML) used jackknife technique to reduce bias in estimating the log-marginal likelihood without "new" information.
>     Although both our approach and Nowozin's paper use linear combination to reduce the bias, our method is more computationally efficient.
>     Specifically, we only require resampling $k$ times, whereas the jackknife method involves a leave-$m$-strategy for $1\le m \le k$ and calculate the average within each $m$ which leads to substantially higher computational cost.
>
> **Q3:** Can the authors provide an intuitive explanation for the form of $D\_{n,k}$? Even after reading through the proof of Theorem 1 a few times, it is difficult to understand why the form $D\_{n,k}$ enables better approximation accuracy than the naive plug-in estimator.
>
> **Answer.** We have provided an intuitive explanation for the form of $D\_{n,k}$ implicitly in Remark 2.
>     To summarize, we view the operator $B\_n f(\pi):=\mathbb{E}[f(\hat{\pi})]$  as a good approximation of $f(\pi)$, i.e., $B\_n\approx I$, where $I$ is the identity operator.
>     This implies that the error operator $E:=I-B$ is a "small" operator.
>     Under this heuristic, if $E f(\pi)=\mathcal{O}(n^{-1})$,
>     intuitively we have $E^k f(\pi)=\mathcal{O}(n^{-k})$.
>     Using the binomial expansion of $E^k=(I-B\_n)^k$, we have $E^k f(\pi)=f(\pi)-\sum\_{j=1}^k\binom{k}{j}(-1)^{j-1}B_n^jf(\pi)=f(\pi)-\mathbb{E}[\sum\_{j=1}^k\binom{k}{j}(-1)^{j-1}B\_n^{j-1}f(\hat{\pi})]=f(\pi)-\mathbb{E}[D\_{n,k}f(\hat{\pi})]$.
>     This representation motivates the specific form of $D\_{n,k}$.
>
>
> **Q4:** Does this method overcome limitations of the empirical Dirac distribution in high dimensional settings?
>
> **Answer.** In our response to Question 1, we have explained how to avoid the curse of dimensionality and included a simple example to illustrate how the sample size $n$ and dimension $d$ may relate.

---

> > ### Comment · Reviewer_yaah · 2025-08-06
> > **Thanks for the detailed respose. I will raise my score but I am still not sure about how scalable the method is.**
> >
> > I appreciate the additional experiments with the more complex distribution and the large values of $k$ for the Gaussian-Bernoulli experiment. These definitely make the paper stronger. On one hand, I am still quite skeptical that the proposed method can overcome the curse of dimensionality and yield practical benefits in typical machine learning scenarios, especially since in the application cases $X$ is simply 1D. On the other hand, I appreciate the novelty and statistical insights in this paper. I will raise my score.

---

> > > ### Author Response · Authors · 2025-08-08
> > >
> > > Thank you for your thoughtful review and for raising the score of our work.
> > >
> > > “The Intrinsic Dimension of Images and Its Impact on Learning” by Pope et al. (2021, ICLR) provides empirical evidence for the widely held view that the success of deep learning on high-resolution data is driven by the underlying low-dimensional structure. They show that common natural image datasets such as MNIST, CIFAR-10 and ImageNet have an intrinsic dimension far smaller than the number of pixels, and that the extrinsic dimension, the dimension of the ambient space in which data is embedded, has little impact on generalization. For example, although each ImageNet image contains 150,528 pixels, its intrinsic dimension is estimated to be only between 26 and 43.
> > >
> > > Similarly, "Debiasing Evidence Approximations: On Importance-weighted Autoencoders and Jackknife Variational Inference" by Nowozin (2018, ICML) has successfully applied jackknife technique to reduce the bias in estimating the log-marginal likelihood to real datasets like MNIST. Their experiment suggests that for real data, there could be an intrinsic dimension much smaller than the ambient dimension, so that the curse of dimensionality may be alleviated.
> > >
> > > Thus, our method has the potential to mitigate the curse of dimensionality and offer practical benefits in many machine learning scenarios.

---

### Decision · Program_Chairs · 2025-09-17

**Decision:**

Accept (poster)

**Comment:**

This paper presents a black-box $k$-th order debiasing framework that combines recursively sampled empirical priors with signed weights to yield an approximate posterior that achieves a bias rate of $\mathcal{O}(n^{-k})$ and variance $\mathcal{O}(n^{-1})$.

Reviewers recognized the solid and elegant theoretical contributions as its primary strength, and the uncertainty around high-dimensional viability as the primary weakness. The rebuttal process was productive, with the authors adding new non-trivial experimental results and other mathematical details requested by reviewers, making the paper stronger.

Overall, the paper makes a good contribution that, despite modest empirical results currently, can lead to more advances in the future. Therefore, I am happy to recommend acceptance. I strongly suggest that the authors follow through on all promises made to update the final paper during the discussion period.